# Low-loss YIG-based magnonic crystals with large tunable bandgaps

Huajun Qin[1], Gert-Jan Both[1,2], Sampo J. Hämäläinen[1], Lide Yao[1] & Sebastiaan van Dijken [1]

Control of spin waves in magnonic crystals is essential for magnon-based computing. Crystals made of ferromagnetic metals offer versatility in band structure design, but strong magnetic damping restricts their transmission efficiency. Yttrium iron garnet (YIG) with ultralow damping is the palpable alternative, yet its small saturation magnetization limits dipolar coupling between discrete units. Here, we experimentally demonstrate low-loss spin-wave manipulation in magnonic crystals of physically separated nanometer-thick YIG stripes. We enhance the transmission of spin waves in allowed minibands by filling the gaps between YIG stripes with CoFeB. Thus-formed magnonic crystals exhibit tunable bandgaps of 50–200 MHz with nearly complete suppression of the spin-wave signal. We also show that Bragg scattering on only two units produces clear frequency gaps in spin-wave transmission spectra. The integration of strong ferromagnets in nanometer-thick YIG-based magnonic crystals provides effective spin-wave manipulation and low-loss propagation, a vital parameter combination for magnonic technologies.

[1] NanoSpin, Department of Applied Physics, Aalto University School of Science, FI-00076 Aalto, Finland. [2] Department of Applied Physics, Eindhoven University of Technology, 5600 MB Eindhoven, Netherlands. Correspondence and requests for materials should be addressed to H.Q. (email: huajun.qin@aalto.fi) or to S.v.D. (email: sebastiaan.van.dijken@aalto.fi)

**B**and structure engineering and low-loss propagation of information-carrying entities are the cornerstones of both electronics and magnonics. Whereas electronics exploits dissipative transfer of electric charges, magnonics is based on low-energy transport of angular momentum in the form of spin waves. Active control over the phase and the amplitude of spin waves offers promising prospects for nano-technologies[1] because their wavelength is tunable down to the nanometer scale. Magnonic crystals provide this quintessential functionality[2–7], with one-dimensional arrays of parallel magnetic stripes being an archetypical example[8–13]. Microwave magnetic fields excite collective spin-wave modes if the stripes are coupled via dipolar magnetic fields. As spin waves propagate through a magnonic crystal with positive group velocity when the magnetization aligns along the stripes, this Damon-Eshbach (DE) configuration is most relevant for magnonic devices. Patterning of ferromagnetic metals such as permalloy (Py), Co, and CoFeB into stripe arrays allows for versatile engineering of magnonic band structures[8–15]. For these materials, DE spin waves are highly dispersive and hence their group velocity is large. Strong dispersion also offers large tuning of the position and width of allowed minibands and forbidden frequency gaps by variation of the stripe width, lattice constant, or applied magnetic field. However, magnonic crystals made of ferromagnetic metals suffer from inefficient transmission because of strong Gilbert damping. The decay length of spin waves in these materials is limited to a few tens of micrometers[7].

Ferrimagnetic YIG exhibits magnetic damping about two orders of magnitude lower than 3d transition metals and is considered as the foremost material for magnonics[1,16]. Research on YIG-based magnonic crystals is extensive, but in contrast to ferromagnetic metals, spin-wave propagation across individual YIG structures is hampered by weak dipolar coupling. This bottleneck, which stems from the small magnetization of YIG, limits the transmission of spin waves. Several alternatives have been investigated such as micrometer-thick YIG films with shallow air grooves[17–21], metallic stripes on top[22,23], current carrying meander structures[24], width modulations[25], and light absorbers[26]. Because spin-wave scattering in these magnonic crystals is weak compared to discrete lattices, many unit cells ($N = 5–35$ in refs. [17–26]) are needed to open up bandgaps. The inefficiency of spin-wave reflection from single lattice units also restricts the bandgap size[27]. In combination with the relatively flat spin-wave dispersion relation of YIG, this has thus far produced small bandgaps ranging from only a few to a few tens of MHz in YIG-based magnonic crystals.

Here, we experimentally demonstrate discrete YIG-based magnonic crystals combining robust tunable bandgaps of up to 200 MHz and low-loss transmission in allowed minibands over a distance of 200 μm. The crystals consist of 260-nm-thick YIG stripes that are separated from each other by only a few ($N = 2–4$) narrow channels of air or CoFeB. Using broadband spin-wave spectroscopy and micromagnetic simulations, we show how the forced precessions of magnetization in CoFeB efficiently enhance the coupling between quasi-confined spin-wave modes in YIG stripes, causing longer distance propagation of DE spin waves at allowed frequencies. In contrast, transmission of spin waves inside the bandgaps is almost completely suppressed. The positions and widths of the bandgaps are tunable by variation of the external bias field, lattice constant, or stripe width. In comparison, the band structure of YIG-based magnonic crystals without CoFeB (airgap-separated stripes) exhibits slightly larger bandgaps, but propagating spin waves damp out more quickly because of reduced dipolar coupling between the YIG stripes.

## Results

**Spin-wave transmission in continuous YIG films.** Single-crystal ferrimagnetic YIG films with a thickness of 260 nm were grown on (111)-oriented $Gd_3Ga_5O_{12}$ (GGG) substrates using pulsed laser deposition (PLD). After growth, the film structure was characterized by X-ray diffraction (XRD) and transmission electron microscopy (TEM). Figure 1a shows a cross-sectional high-resolution TEM image and the corresponding diffraction pattern of an as-grown YIG film on GGG. The well-ordered atomic structure and sharp diffraction spots illustrate coherent epitaxial growth of YIG(111) on GGG(111). Magnetic damping in the YIG films was measured using a vector network analyzer ferro-magnetic resonance (VNA-FMR) setup. Figure 1b presents the linewidth of VNA-FMR spectra as a function of resonance frequency. From a linear fit to the data using $\Delta f = 2\alpha f + v_g \Delta k$[28], where $v_g$ is the group velocity and $\Delta k$ is the excitation spectrum width of the antenna, we find a Gilbert damping constant of $\alpha = (4.4 \pm 1.5) \times 10^{-4}$. Spin-wave transmission in the YIG films was studied using broadband spin-wave spectroscopy. Figure 1c shows a schematic of the measurement geometry. Two 6-μm-wide antennas made of 3 nm Ta/200 nm Au were patterned on the YIG films with a separation of $x = 200$ μm. Spin waves were excited by spatially inhomogeneous microwave magnetic fields from the first antenna and inductively detected by the second antenna using a vector network analyzer. The spin waves had wave vectors ranging from zero to about $\pi/w$, where $w = 6$ μm corresponds to the antenna width[29]. We characterized spin-wave transmission by recording the $S_{12}$ scattering parameter.

Figure 1d shows a typical transmission spectrum (imaginary part of $S_{12}$) of a 260-nm-thick YIG film at an external magnetic bias field of 5 mT, which suffices to fully saturate the magnetization. An oscillating spin-wave signal is measured above the FMR frequency of ~1.0 GHz. The strong oscillations indicate efficient spin-wave propagation between the two antennas. The group velocity of the spin waves ($v_g$) is extracted from the frequency separation ($\delta f$, indicated in Fig. 1d) using $v_g = \delta f \times x$[30,31]. For frequencies ranging from 1.1 to 1.8 GHz, we find a gradual decrease of the group velocity from 11 to 5.5 km s$^{-1}$ (upper panel in Fig. 1e). This method assumes the spin-wave dispersion relation to be linear. The small non-linearity of the actual dispersion curve produces an estimated error of 4% in the group velocity values. The lowering of $v_g$ with frequency is explained by a flattening of the DE spin-wave dispersion relation with increasing wave vector. The large group velocity in our 260-nm-thick YIG films enables transport of spin waves over long distances. Using $l_d = v_g/2\pi\alpha f$, we extract a spin-wave decay length ($l_d$) of up to 4 mm in a small magnetic field of 5 mT (lower panel in Fig. 1e). This low-loss transmission provides a solid platform for the design of magnonic crystals. Figure 1f shows the imaginary part of $S_{12}$ for different external magnetic bias fields. Asymmetric spin-wave intensities for negative and positive bias fields indicate a non-reciprocal effect, caused by dissimilar DE spin-wave propagation at the surface and interface of the YIG film[32–35].

**Spin-wave transmission in YIG-based magnonic crystals.** Magnonic crystals were fabricated by photolithography and argon-ion milling of periodic grooves in 260-nm-thick YIG films. The grooves were centered between the two microwave antennas and extended all the way to the GGG substrate. In addition to airgap-separated YIG stripes, we also fabricated bicomponent magnonic crystals by filling the grooves with 260 nm CoFeB using magnetron sputtering. A large number of magnonic crystals with different combinations of groove width ($w$) and lattice constant ($a$) were patterned. Figure 2a shows a schematic of our magnonic crystal and measured height profiles of samples without

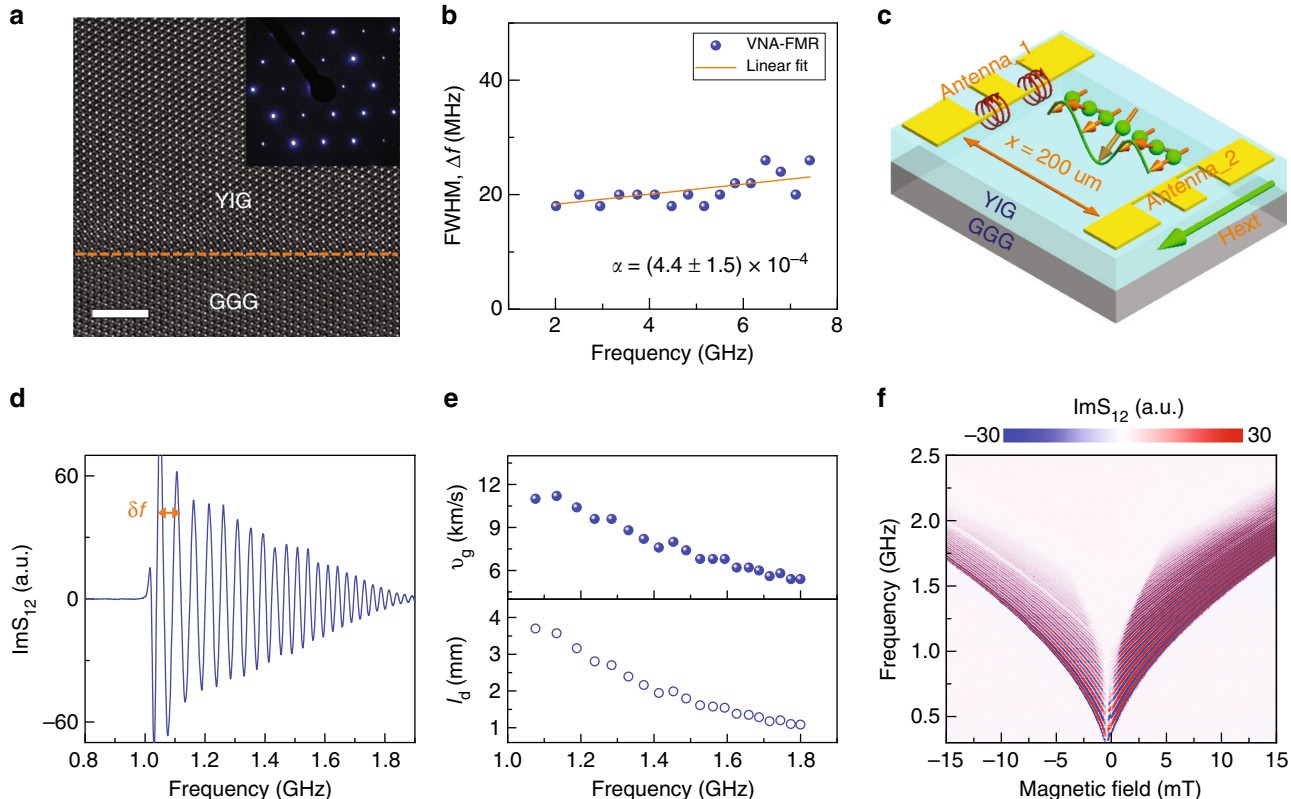

**Fig. 1** Properties of continuous nanometer-thick YIG films. **a** High-resolution TEM image and corresponding diffraction pattern of a 260-nm-thick YIG film. The dashed line indicates a sharp interface between the GGG(111) substrate and single-crystal YIG film. The white scale bar corresponds to 5 nm. **b** Linewidth of VNA-FMR spectra as a function of frequency. The data are obtained by variation of the external magnetic bias field. **c** Schematic of the broadband spin-wave spectroscopy measurement geometry. **d** Spin-wave transmission spectrum (imaginary part of $S_{12}$) for a continuous 260-nm-thick YIG film at a magnetic bias field of 5 mT. **e** Spin-wave group velocity ($v_g$) and decay length ($l_d$) in YIG. **f** Imaginary part of $S_{12}$ at different external magnetic bias fields

(orange line) and with (green line) CoFeB in the grooves. During broadband spin-wave spectroscopy, spin waves are excited in the continuous part of the YIG film near the first antenna. These spin waves propagate perpendicular to the stripes while a small external bias field ($H_{ext}$) establishes the DE configuration (Fig. 2a). Figure 2b shows the imaginary part (top panel) and amplitude (bottom panel) of the $S_{12}$ scattering parameter as a function of frequency for a magnonic crystal with $N = 4$ CoFeB-filled grooves, $a = 30$ μm, and $w = 2.5$ μm (hereafter referred to as $a30w2.5$). The bias field in this measurement is 5 mT. Four allowed minibands and three forbidden frequency gaps of >100 MHz are observed. Transmission of spin waves in the bandgaps is almost completely suppressed, as illustrated by a reduction of the signal amplitude to the background level measured at $f < 1$ GHz. In the allowed minibands, spin waves propagate efficiently through the bicomponent magnonic crystal. The spin-wave signal of the CoFeB/YIG stripe array is only a factor ~3 smaller than that of an unpatterned YIG film (blue line in Fig. 2b). From the frequency separation between maxima and minima in the $ImS_{12}$ signal near the center of the minibands, we derive a spin-wave group velocity ranging from 11 km s$^{-1}$ in the first band (~1.20 GHz) to 4.4 km s$^{-1}$ in the fourth band (~1.73 GHz). These group velocities are similar to those measured on a continuous YIG film (Fig. 1e).

Figure 2c shows spectra of the imaginary part of $S_{12}$ at different external magnetic bias fields. In our magnonic crystal, large bandgaps persist up to tens of mT. The forbidden gaps are largest near zero bias field and gradually diminish with increasing field strength. We note that magnetic switching in the YIG and CoFeB

stripes occurs independently at a field of about 0.1 and 2.0 mT, respectively (Supplementary Fig. 1). An antiparallel magnetization configuration is therefore attained in small bias fields. Judging by the continuous curves of Fig. 2c, switching to a parallel magnetization state at −2.0 or +2.0 mT does not affect the spin-wave transmission signal. Reprogramming of the magnonic band structure by toggling between differently aligned magnetization states, as demonstrated for discrete Py wires[11], does therefore not occur in our bicomponent magnonic crystals. We attribute this invariance to the large difference in stripe width and the relatively small precession of magnetization in CoFeB at 1.0–1.8 GHz. Figure 2d presents the angular dependence of spin-wave transmission for a 5 mT field. In this graph, a field angle of $\phi_H = 90°$ corresponds to the DE geometry ($M \parallel$ stripes) and $\phi_H = 0°$ or 180° would represent results for backward volume magnetostatic spin waves (BVMSWs, $M \perp$ stripes). Three bandgaps are resolved for angles between 60° and 120°. Rotation of the field away from the DE configuration reduces the signal intensity, bandgap positions, and gap widths (the angular dependence of the latter parameter is more clearly illustrated in Supplementary Fig. 2 for a $a50w5$ crystal). These effects are explained by a weakening of the excitation efficiency and a flattening of the dispersion relation upon rotation of magnetization toward the spin-wave wave vector[31,36]. We prepared a large number of bicomponent magnonic crystals by varying parameters $a$ and $w$ in 1 and 0.25 μm steps. Figure 2e, f summarize the transmission of spin waves in all these crystals. Using only four CoFeB stripes, we observe up to seven bandgaps and strong tuning of bandgap parameters with $a$. While the CoFeB stripe

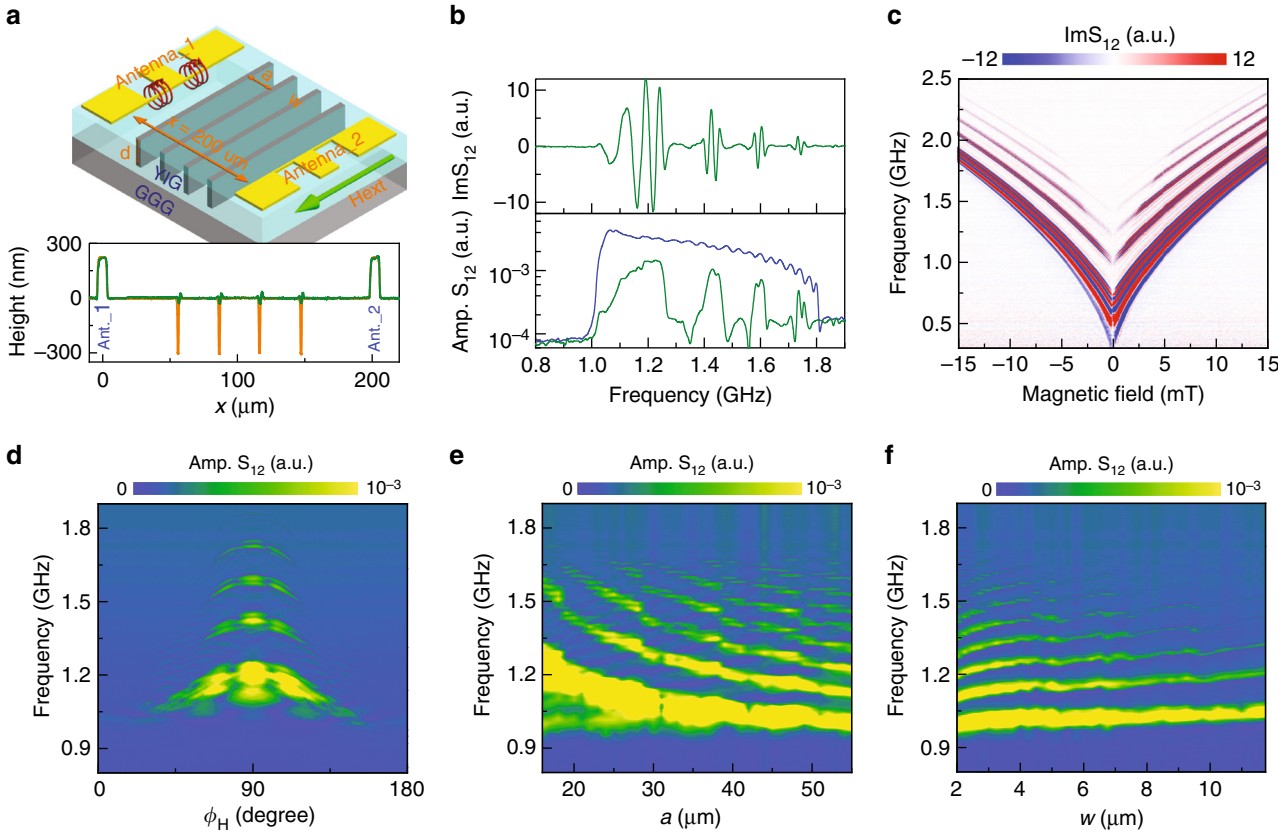

**Fig. 2** Spin-wave transmission in nanometer-thick YIG-based magnonic crystals. **a** Schematic of a one-dimensional magnonic crystal. The crystal consists of physically separated YIG stripes with a thickness of 260 nm and narrow air grooves or grooves that are filled by CoFeB. Two 6-μm-wide microwave antennas are patterned on top of YIG. Their separation is $x = 200$ μm. The bottom graph shows height profiles for magnonic crystals with air grooves (orange line) and grooves that are filled by CoFeB (green line). **b** Spin-wave transmission spectra (imaginary part and amplitude of $S_{12}$) for a a30w2.5 CoFeB/YIG magnonic crystal at a magnetic bias field of 5 mT. The reference spectrum (blue line) is recorded on a continuous YIG film. **c** Imaginary part of $S_{12}$ for the same crystal at different magnetic bias fields. **d** Dependence of spin-wave transmission spectra (amplitude of $S_{12}$) on magnetic field angle in a a30w2.5 CoFeB/YIG crystal. In this graph, 90° corresponds to the excitation of DE spin waves. **e, f** $S_{12}$ amplitude as a function of **e** lattice constant and **f** CoFeB stripe width. In **e**, $w = 2.5$ μm. In **f**, $a = 50$ μm. All the data are recorded on magnonic crystals with $N = 4$ grooves

width affects the magnonic band structure less, the efficiency of spin-wave transmission reduces quickly when $w$ becomes large.

To gauge the effect of CoFeB on the performance of our magnonic crystals in more detail, we directly compare spin-wave transmission spectra of bicomponent lattices to those of identical YIG stripe arrays without CoFeB. Figure 3 presents results for three different magnonic crystals, a30w2.5, a30w5, and a50w5, with orange and green lines representing data for pure YIG and CoFeB/YIG lattices, respectively. Whereas both types of crystals show robust bandgaps, their widths are slightly larger for stripe arrays with air grooves. Moreover, the spin-wave signal decays faster in crystals made of pure YIG, which is particularly apparent for $w = 5$ μm. Filling of the air grooves with CoFeB thus enhances the efficiency of spin-wave transmission in nanometer-thick YIG-based magnonic crystals.

We quantify the bandgap size and efficiency of spin-wave transmission by depicting their dependence on parameter $w$ in Fig. 4. The spin-wave transmission spectra of Fig. 4a–c compare results for pure YIG (orange lines) and CoFeB/YIG (green lines) lattices with $a = 30$ μm and $w = 2$, 4, and 6 μm. The blue lines in these graphs are measured on a continuous area of the same YIG film. From these data (and results on lattices with $w = 3$ and 5 μm), we extracted the spin-wave transmission signal in the first three minibands relative to the $S_{12}$ amplitude measured on continuous YIG (Fig. 4d). The results demonstrate that the efficiency of spin-wave transmission drops 16 and 22 times in

the 2nd and 3rd miniband if the air-groove width is enlarged from 2 to 6 μm. In contrast, the $S_{12}$ signal only reduces by a factor <3 upon an increase of $w$ if the grooves are filled with CoFeB. Because of these dissimilar dependencies, the transmission of spin waves in higher order minibands of CoFeB/YIG lattices is at least one order of magnitude larger than in crystals made of pure YIG if $w > 5$ μm. Figure 4e shows the dependence of bandgap size on groove width. The bandgap increase slightly with $w$, but the difference between YIG and CoFeB/YIG magnonic crystals remains small (<15%).

The center frequencies of the bandgaps are accurately mapped onto the spin-wave dispersion relations of the magnonic crystals at wave vectors that satisfy conditions of Bragg reflection. To illustrate this, we plot the dispersion curves for DE spin waves in a30w2.5, a30w5, and a50w5 CoFeB/YIG magnonic crystals in the lower panels of Fig. 3. The dispersion relations were calculated using[37]

$$f = \frac{\gamma \mu_0}{2\pi} \sqrt{(H_{ext} + H_{ani})(H_{ext} + H_{ani} + M_s) + \frac{M_s^2}{4}(1 - e^{-2kd})}.$$

(1)

As input parameters, we used $\frac{\gamma}{2\pi} = 28$ GHz T$^{-1}$, $d = 260$ nm, $\mu_0 H_{ext} = 5$ mT, magnetic anisotropy field $\mu_0 H_{ani} = 1.5$ mT, and effective saturation magnetizations $M_s = 185$ kA m$^{-1}$ (for a30w2.5 and a50w5) and $M_s = 192$ kA m$^{-1}$ (for a30w5). The values of $\mu_0 H_{ani}$ and $M_s$ were extracted from the dependence

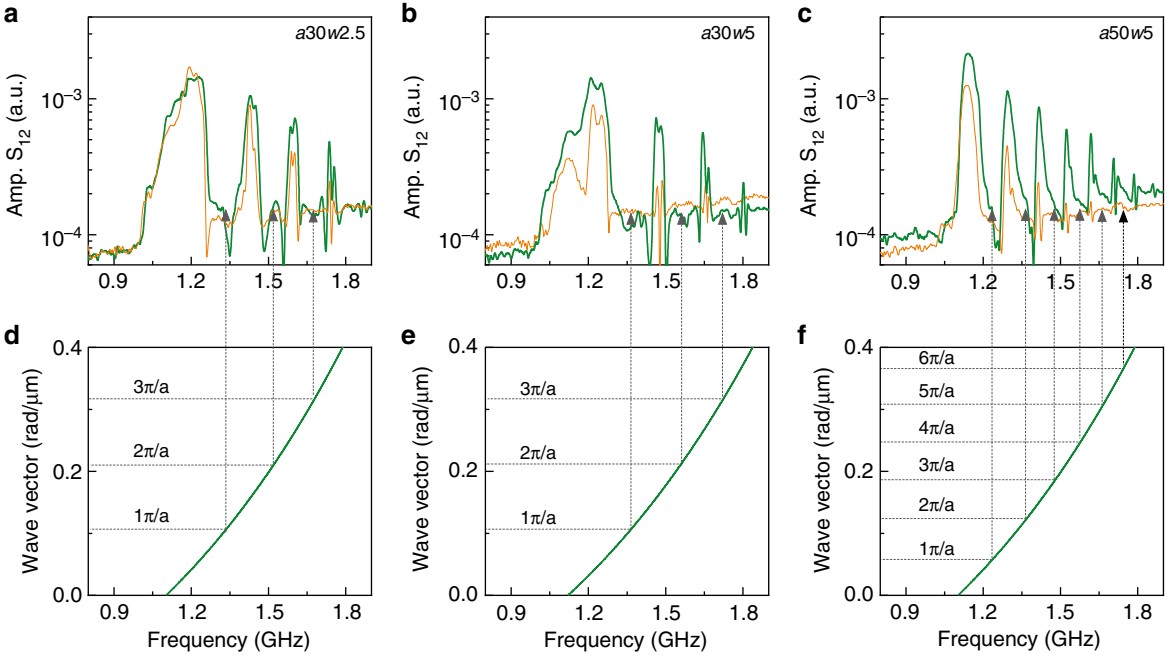

**Fig. 3** Spin-wave dispersion relations and Bragg reflection. **a–c** Transmission spectra of CoFeB/YIG bicomponent magnonic crystals (green lines) and YIG stripes with airgaps (orange lines). The magnonic crystals consist of $N = 4$ grooves. The lattice constants and groove widths are indicated in the graphs. **d–f** Calculated dispersion relations of DE spin waves for the bicomponent magnonic crystals. The vertical and horizontal dashed lines show the center frequencies of bandgaps and corresponding wave vectors

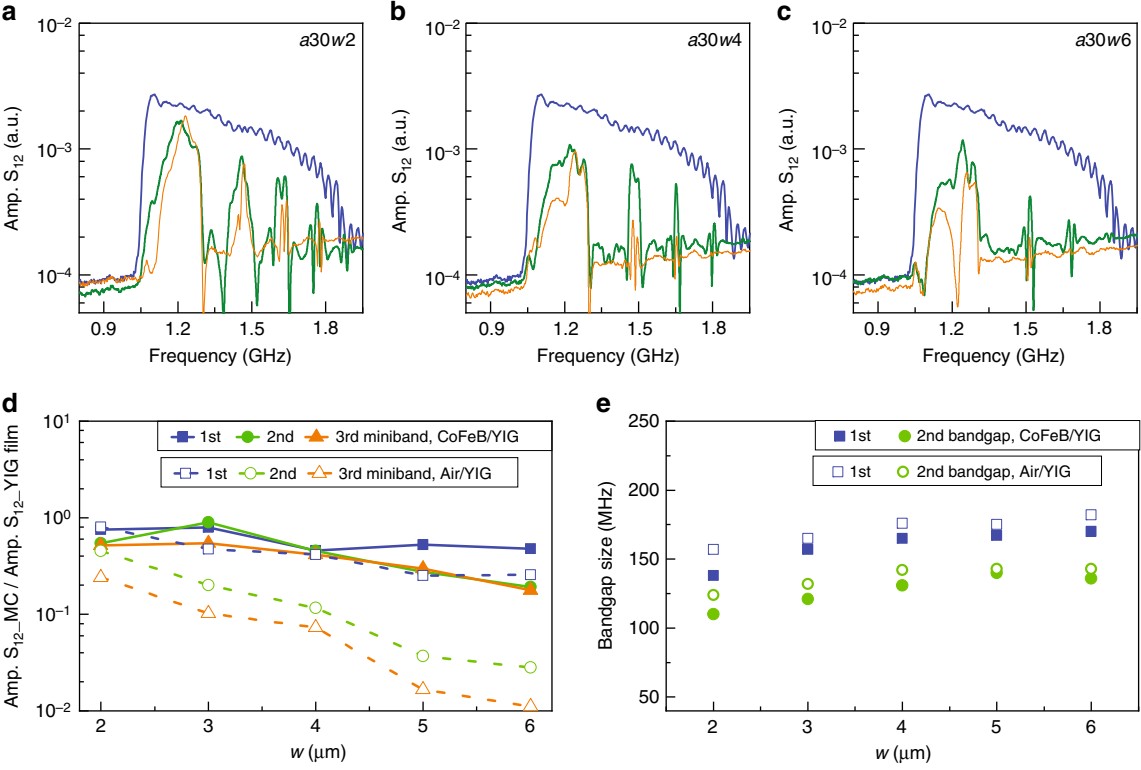

**Fig. 4** Dependence of spin-wave transmission and bandgap size on groove width. **a–c** Transmission spectra of pure YIG (orange lines) and CoFeB/YIG (green lines) magnonic crystals with $N = 4$ grooves. The blue line depicts a reference spectrum measured on a continuous part of the same YIG film. **d** Dependence of spin-wave transmission on groove width $w$ in the first three minibands. The amplitudes of the $S_{12}$ scattering parameter in the magnonic crystals (Amp. $S_{12}\_MC$) are compared to those recorded on continuous YIG (Amp. $S_{12}\_YIG$ film). **e** Variation of the bandgap size with groove width

of FMR frequency on external magnetic field (Supplementary Fig. 3). These measurement data were obtained by recording the $S_{12}$ scattering parameter in transmission on the same magnonic crystals. The slight increase of $M_s$ for the a30w5 crystal is attributed to the larger CoFeB/YIG ratio[14,38]. For all crystals, the center frequencies of the bandgaps correspond to wave vectors $n\pi/a$, where $n$ is the order number of the bandgap and $\pi/a$ is the first Brillouin zone boundary. Efficient Bragg reflection of incoming spin waves in our magnonic crystals is thus responsible for the opening of robust bandgaps with high rejection efficiencies.

We note the occurrence of an extra feature at ~1.2 GHz in the lowest frequency pass band of the a30w4, a30w5, and a30w6 crystals (Figs. 3b and 4b, c). This feature, which signifies a real suppression of the spin-wave transmission signal, appears more clearly if the grooves become wider. Moreover, the suppression is stronger if the YIG stripes are separated by air rather than CoFeB. At the feature frequency, the wavelength of excited spin waves matches the size of the magnonic crystal (i.e., the distance between the first and last groove). For more details see Supplementary Fig. 4.

The transmission spectra presented thus far are recorded on magnonic crystals with $N = 4$ grooves. This limited number already suffices for the formation of deep bandgaps. In previously studied magnonic crystals, a larger number of spin-wave reflectors is often required to significantly suppress the transmission of spin waves in the frequency gaps[5–7]. To test whether the number of grooves in our nanometer-thick YIG-based magnonic crystals can be reduced even further, we fabricated samples with $N = 2, 3 .. 6$. Figure 5 shows results for CoFeB/YIG samples with $a = 30\,\mu m$ and $w = 2.5\,\mu m$. The experiments demonstrate that the magnonic crystal with only two grooves already exhibits pronounced bandgaps. With increasing N, the bandgaps deepen and the efficiency of spin-wave transmission in the allowed minibands decreases. The width of the bandgap does not depend strongly on N, which is in line with previous observations that this parameter is mainly defined by the reflection efficiency of a single reflector[27].

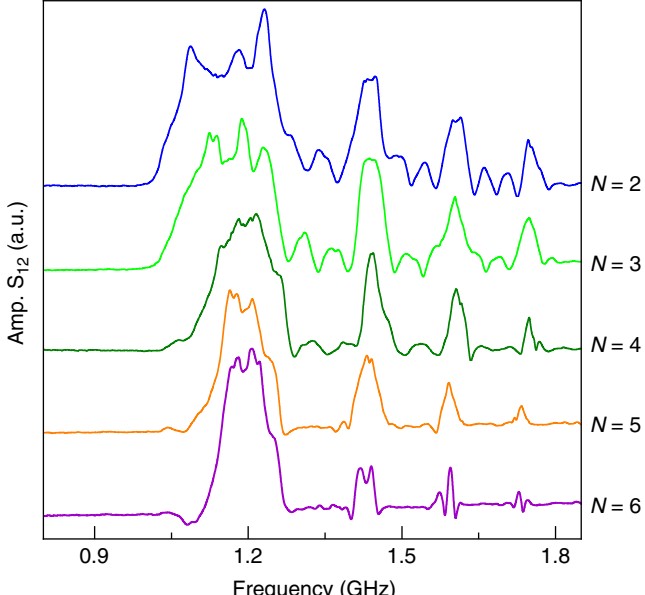

**Fig. 5** Dependence of spin-wave transmission spectra on the number of grooves. Transmission spectra of CoFeB/YIG bicomponent magnonic crystals with $w = 2.5\,\mu m$, $a = 30\,\mu m$, and different numbers of CoFeB-filled grooves (N). For clarity, the spectra are plotted with constant offsets

**Simulations of spin-wave profiles and transmission spectra.** We used the MuMax3 micromagnetic solver[39] to simulate spin waves in YIG and CoFeB/YIG magnonic crystals. The simulation area consisted of a 655-μm-long and 260-nm-thick YIG film with N periodic airgaps or CoFeB stripes (Fig. 6a). The structure was discretized using finite-difference cells of size $x = 40\,nm$, $y = 40\,nm$ and $z = 32.5\,nm$. One-dimensional periodic boundary conditions were applied along the $y$-axis to mimic a lattice with infinitely long stripes. To reduce reflections from the edges along $x$, we added two 5-μm-wide areas with a high Gilbert damping constant of $\alpha = 0.5$. As input parameters, we used $M_s = 185\,kA\,m^{-1}$ (YIG), $M_s = 1150\,kA\,m^{-1}$ (CoFeB), $A_{ex} = 3.1\,pJ\,m^{-1}$ (YIG), $A_{ex} = 16\,pJ\,m^{-1}$ (CoFeB), $\alpha = 0.003$ (YIG) and $\alpha = 0.005$ (CoFeB). For YIG, the relatively large damping parameter was intentionally selected to limit the computation time and reduce reflections from the edges of the simulation area. Moreover, we added a 0.16-μm-wide non-magnetic layer between the YIG and CoFeB stripes. During the course of our simulation study, we noted that good agreements with experimental spin-wave transmission spectra could only be attained with the inclusion of a thin non-magnetic spacer layer. We attribute the formation of this layer to argon-ion milling in our lithography process. Complex oxides are particularly sensitive to ion-beam milling and, therefore, we anticipate YIG to be structurally and magnetically modified at the groove edges. Supplementary Figure 5 shows how the thickness of the non-magnetic layer influences the transmission spectra. We used sinc-function-type field pulses with a cutoff frequency of 3 GHz to generate spin-wave intensity maps and sinusoidal ac magnetic fields with fixed frequencies to determine the time evolution of spin-wave profiles. The fields were applied along the $x$-axis over a 6-μm-wide area in the continuous YIG part. To visualize the propagation of spin waves in magnonic crystals, we analyzed the $x$-component of magnetization ($m_x$), whose dynamic response is much stronger than $m_z$ because of the strong demagnetization field in our thin-film geometry.

Figure 6a shows the spatial distribution of spin-wave intensity in a bicomponent a30w2.5 magnonic crystal. In this simulation, spin waves with frequencies ranging from 1.0 to 2.5 GHz are excited near $x = 0\,\mu m$. Propagating spin waves with a frequency of ~1.35 GHz and wave vectors $k \approx \pi/a$ are efficiently reflected by the one-dimensional stripe array. Consequently, the spin-wave intensity in subsequent YIG stripes drops quickly and transmission across the entire magnonic crystal is almost completely suppressed. Bandgaps are formed also when conditions for higher-order Bragg reflections are met ($k = n\pi/a$). The positions and widths of the simulated bandgaps correspond closely to our experimental data, as illustrated by the transmission spectra on the right side of Fig. 6a. The simulated spectrum (green solid line) was obtained by averaging $m_x$ over a 6-μm-wide area at $x = 200\,\mu m$. Apart from a stronger decay of spin waves in the simulations (caused by the use of $\alpha = 0.003$ for YIG), the bandgap parameters agree well. In the allowed minibands between the frequency gaps, spin waves efficiently propagate through the bicomponent magnonic crystal. Nodes in the spin-wave intensity map of Fig. 6a demonstrate that the spin waves are partially confined in the YIG stripes. These quasi-standing modes couple to each other via dynamic dipolar fields. As the dynamic coupling field between neighboring stripes decreases monotonically with node number[5,10], the transmission signal is largest in the first miniband.

Figure 6b depicts the time evolution of spin-wave transmission through a a30w2.5 CoFeB/YIG lattice at a frequency of 1.44 GHz, which corresponds to the second miniband center. After the onset of continuous excitation at $t = 0\,s$, spin waves propagate toward the magnonic crystal. Dynamic dipolar coupling between the YIG stripes results in an efficient perpetuation of the

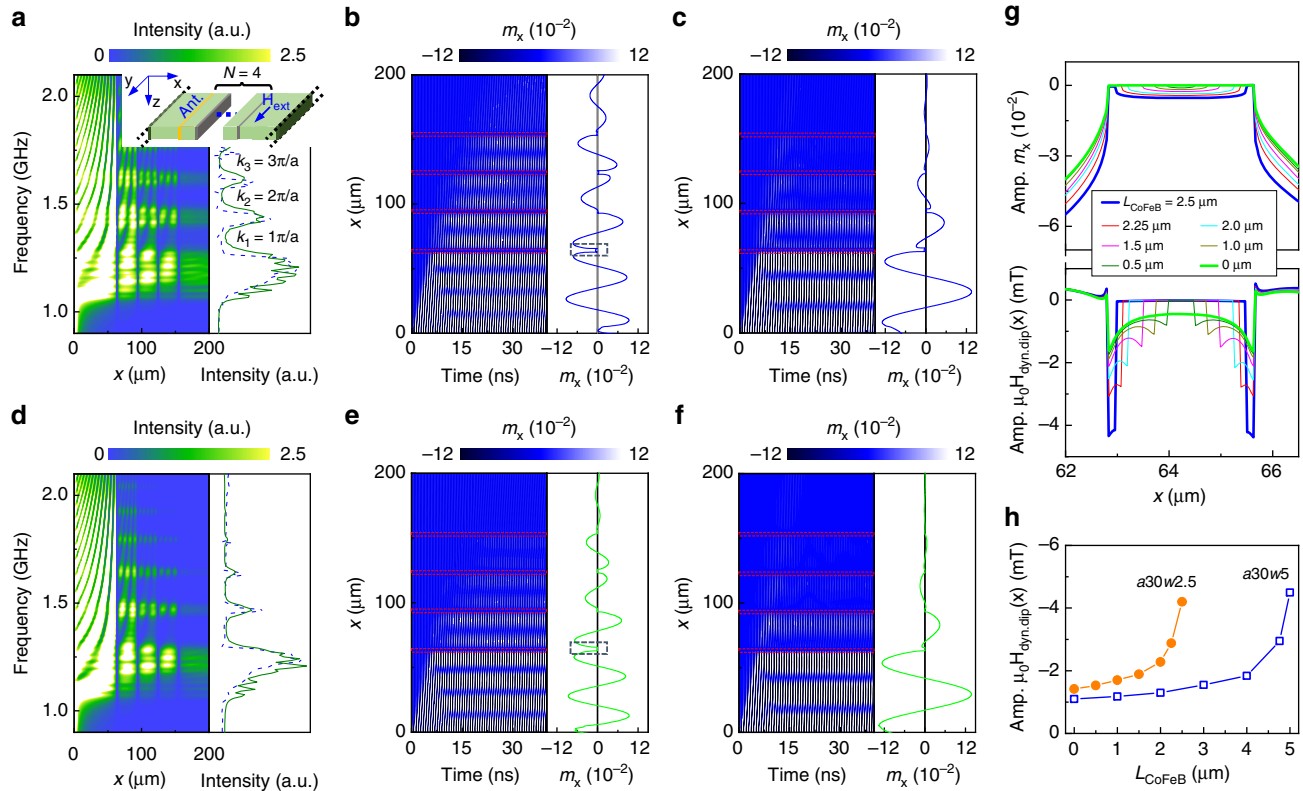

**Fig. 6** Micromagnetic simulations. **a** Spatial distribution of spin-wave intensity in a $a30w2.5$ bicomponent magnonic crystal with $N = 4$ CoFeB stripes. The right panel compares simulated (solid green line) and measured (dashed blue line) transmission spectra. The inset illustrates the simulation geometry. **b**, **c** Time evolution of spin-wave profiles during ac excitation with a sinusoidal magnetic field at a fixed frequency of **b** 1.44 GHz and **c** 1.35 GHz. Spin-wave profiles after an excitation time of 30 ns are shown on the right side of the graphs. The system reached steady-state excitation at this time. **d–f** Similar data as in **a–c**, but for a $a30w2.5$ crystal without CoFeB, i.e., discrete YIG stripes separated by airgaps. **g** Amplitude of $m_x$ near the first groove (top panel). The bold blue and green lines correspond to data recorded in the boxes of **b**, **e**. The curves show data for a fixed groove width of $w = 2.5 \, \mu m$, but different amounts of CoFeB ($L_{CoFeB}$). The bottom panel of **g** shows the amplitude of the $x$-component of the dynamic dipolar field for different amounts of CoFeB. **h** Variation of the dynamic dipolar field near the edge of the first groove ($x = 62.9 \, \mu m$ in **g**) as a function of $L_{CoFeB}$ for $a30w2.5$ and $a30w5$ magnonic crystals

excitation across the crystal. The phase of spin waves in neighboring YIG stripes is preserved, illustrating the collective character of the modes. In Fig. 6c, we show a similar simulation at 1.35 GHz. Because this frequency is located in the first bandgap, spin waves are effectively reflected by the lattice and their propagation is limited to 1–2 YIG stripes.

For comparison, we now present simulations for the same $a30w2.5$ crystal, but with CoFeB being replaced by air. Figure 6d–f summarizes the results. In analogy to the bicomponent crystal, several allowed minibands and bandgaps are simulated. Based on these data, we conclude that the separating material (CoFeB versus air) has two main effects. First, bandgaps of the pure YIG magnonic crystal are a bit wider than those of the CoFeB/YIG lattice. Second, transmission of spin waves is more efficient if the air grooves are filled by CoFeB. Both observations agree with the experimental data of Figs. 3 and 4.

The simulations in Fig. 6b, c indicate that magnetization precessions in the narrow CoFeB stripes are much smaller than in YIG. To quantify the effect of CoFeB on dynamic coupling between the YIG stripes, we performed micromagnetic simulations on $a30w2.5$ and $a30w5$ crystals (Fig. 6g, h) with varying amounts of CoFeB. Starting from a pure airgap of fixed width, we positioned narrow CoFeB stripes in the groove center. The width of CoFeB stripe ($L_{CoFeB}$) was varied in steps from 0 (pure airgap) to $w$ (groove fully filled with CoFeB). Figure 6g summarizes the magnetization precession amplitude and $x$-component of the

dynamic dipolar field near the first groove of the $a30w2.5$ crystal. The data are simulated at a frequency that corresponds to the center of the second allowed miniband. Despite the relative small value of $m_x$ in the CoFeB stripe, the use of CoFeB substantially enhances the dynamic dipolar field. As a result, the YIG stripes are coupled more strongly and the efficiency of spin-wave transmission is enhanced compared to the same magnonic crystal with airgaps. Figure 6h summarizes the dependence of the dynamic dipolar field on the amount of CoFeB in the first groove of $a30w2.5$ and $a30w5$ lattices.

Finally, we describe simulations on spin-wave transmission in magnonic crystals with only $N = 2$ grooves. Figure 7 compares results for $a30w2.5$ crystals with CoFeB stripes and airgaps. The simulations confirm that only two spin-wave reflectors suffice for the formation of bandgaps and allowed minibands in discrete magnonic crystals of nanometer-thick YIG, in agreement with experimental data (Fig. 5). Again, the transmission of spin-waves in the allowed minibands is larger for the bicomponent CoFeB/ YIG lattice, while air grooves produce slightly more pronounced bandgaps.

## Discussion

Our results demonstrate that robust bandgaps and high transmission efficiency can be achieved in magnonic crystals based on discrete YIG stripes. YIG films with patterned air grooves have been considered before[17–21]. In these studies, the thickness of

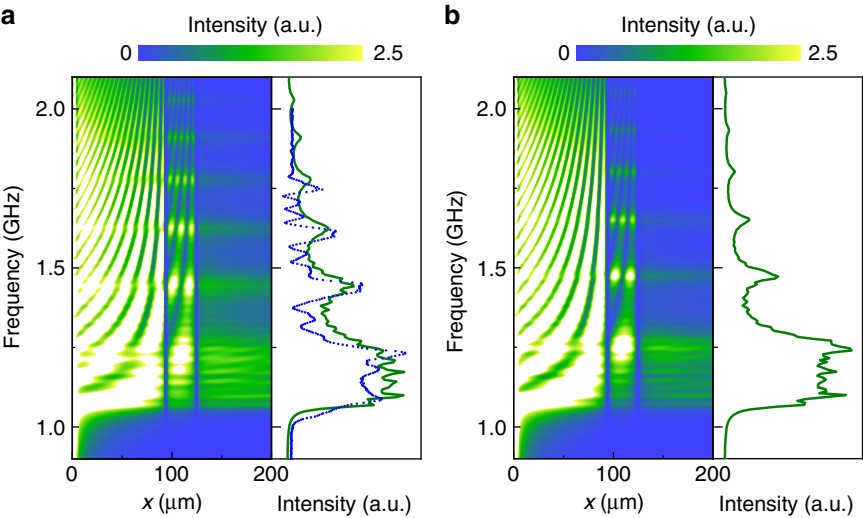

**Fig. 7** Simulations for discrete magnonic crystals with only two units. **a, b** Spatial distribution of spin-wave intensity in a *a*30*w*2.5 crystal with **a** *N* = 2 CoFeB stripes and **b** *N* = 2 airgaps. The right panels show simulated transmission spectra (solid green lines) and an experimental measurement (dashed blue line) for the bicomponent lattice

YIG ranged from a few to several tens of micrometers and grooves did not extend throughout the film. Besides technical challenges of patterning deep narrow grooves in thick YIG films, a strong increase of transmission losses with groove depth justifies the omission of stripe arrays on this thickness scale[17,18]. For micrometer-thick YIG stripes, spin-wave propagation is hampered because in-plane ($m_x$) and out-of-plane ($m_z$) magnetization precessions produce comparable dipolar coupling fields. The combination of these dynamic fields, with one favoring in-phase and the other forcing out-of-phase oscillations, suppresses the spin-wave signal. In our nanometer-thick YIG-based magnonic crystals, the situation is different. Here, the demagnetization field reduces $m_z$ and, hence, stripes predominantly interact via in-plane dynamic dipolar fields. Downscaling the thickness of YIG-based magnonic crystals, which is now possible because of recent advancements in high-quality YIG film growth[40–44], thus provides new opportunities for low-loss manipulation of spin waves in magnonic devices.

Besides the positive impact of thickness reduction, our experimental and simulation data show that spin-wave propagation improves further if the air grooves between YIG stripes are filled by CoFeB. We explain this effect by an enhancement of dynamic dipolar coupling via forced magnetization precessions in the narrow CoFeB stripes (Fig. 6g, h). Despite their small amplitude, these precessions produce a significant dipolar field because the magnetization of CoFeB is large ($M_{s,CoFeB} \approx 8M_{s,YIG}$). The dynamic field that the CoFeB stripes produce adds to the dipolar field from quasi-standing modes in the YIG stripes and, consequently, it promotes spin-wave propagation across the magnonic crystal. Enhanced stripe-to-stripe coupling also affects the band structure. Near-perfect confinement of spin-wave modes in the stripes of a one-dimensional crystal with parameters *w* (gap width) and *a* (lattice constant) would in theory produce narrow minibands for spin waves with $k = n\pi/(a - w)$ and these bands would be separated by large frequency gaps. Dynamic dipolar coupling between stripes reduces the confinement of spin-wave modes. For constant *w* and *a*, an increase of the coupling strength would thus enhance the miniband width and reduce the bandgap size. This is exactly what we observe in our experiments (Figs. 3 and 4) and simulations (Figs. 6 and 7) when the air grooves of our YIG-based magnonic crystals are filled by CoFeB.

The opening of clear bandgaps down to *N* = 2 in our fully discrete nanometer-thick YIG-based magnonic crystals is unique. To illustrate the effect of groove depth on the magnonic band structure, we performed micromagnetic simulations on lattices with *N* = 2 air grooves that are 12.5 or 50% deep (Supplementary Fig. 6a). In both cases, the two grooves do not open up bandgaps in the spin-wave transmission spectra. Bandgaps with only two scatterers are attained if spin waves are efficiently reflected by grooves that fully extend throughout the YIG film. Compared to magnonic crystals made of ferromagnetic metals, our YIG-based magnonic crystals have the advantage of much lower magnetic damping. Because of this, multiple reflections on periodic scatterers are more readily attained, producing stronger intensity modulations in spin-wave transmission spectra. Simulations on crystals with *N* = 2 and damping parameters $\alpha = 0.003$ or $\alpha = 0.01$ (Supplementary Fig. 6b) demonstrate that low magnetic damping is essential for versatile band structure engineering in lattices with small *N*.

Finally, we discuss tuning of the magnonic band structure. Figure 8 summarizes how parameters of the first two bandgaps depend on external magnetic bias field, lattice constant, and CoFeB stripe width. Large tuning of the bandgap frequency (1.2 GHz → 3.4 GHz) and gap size (140 MHz → 50 MHz) is measured and simulated for a *a*30*w*2.5 crystal if the bias field is increased from 3 to 45 mT (Fig. 8a, b). The monotonic decrease of the bandgap width is explained by a flattening of the spin-wave dispersion curve at large magnetic fields[23]. Geometrical parameters affect the band structure via a change of the Brillouin zone boundary ($\pi/a$) or shift in the spin-wave dispersion relation. The latter effect originates from a dependence of the effective magnetization on CoFeB/YIG ratio. Because spin-wave modes are located mostly in the YIG stripes, the frequency and size of the bandgap vary little with CoFeB stripe width (Fig. 8e, f). The band structure is affected more by the lattice constant. A shift of the Brillouin zone boundary to larger wave vectors enhances the frequency and size of the bandgap for small *a* (Fig. 8c, d)). Based on the parameter space depicted in Fig. 8, we conclude that nanometer-thick YIG-based magnonic crystals allow for large tunable bandgaps of 50–200 MHz.

In summary, we have demonstrated operation of fully discrete YIG-based magnonic crystals for the first time. Low-loss

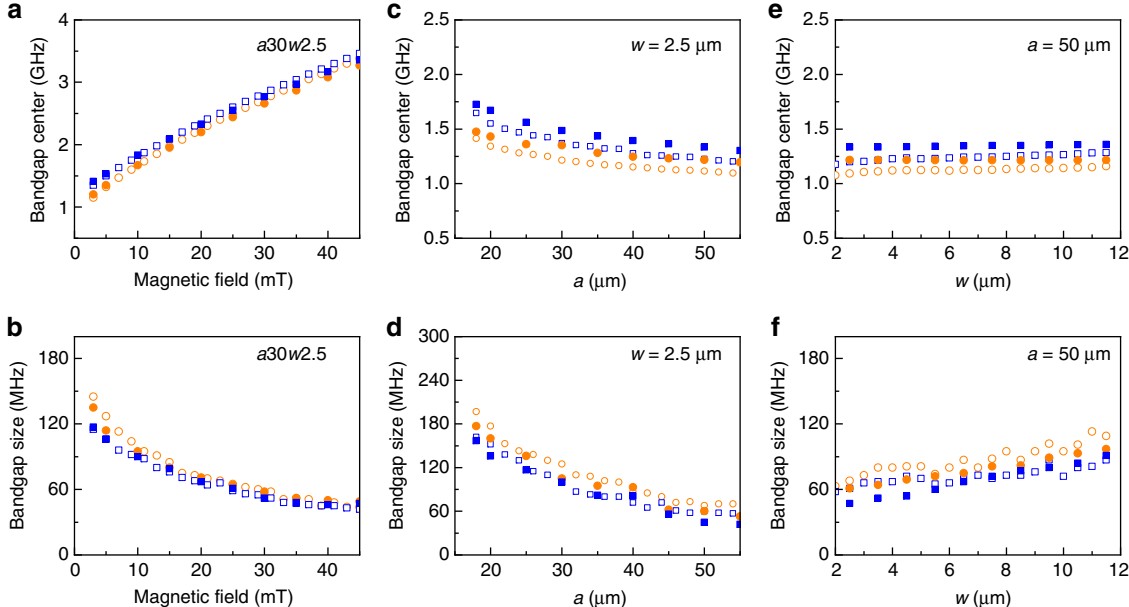

**Fig. 8** Magnonic band structure tuning. **a–f** Measured (empty symbols) and simulated (solid symbols) positions and widths of the first (circles) and second (squares) bandgaps of bicomponent CoFeB/YIG magnonic crystals as a function of **a**, **b** magnetic bias field, **c**, **d** lattice constant, and **e**, **f** CoFeB stripe width

transmission in combination with robust spin-wave manipulation is achieved by limiting the thickness of YIG stripes to the nanometer scale. Because the demagnetization field in our thin crystals suppress magnetization precession in the out-of-plane direction, the YIG stripes effectively interact via dynamic dipolar fields in the film plane. Dynamic magnetic coupling between the YIG stripes is further enhanced if the air grooves that separate them are filled by CoFeB. Incorporation of CoFeB increases the transmission of spin waves in allowed minibands and slightly reduces the size and depth of the bandgaps. The combination of efficient Bragg scattering on airgaps or narrow CoFeB stripes and low magnetic damping in YIG already produces clear bandgaps for the absolute minimum of two scattering units. Bandgap sizes are tunable from 50 to 200 MHz by variation of the lattice constant and groove width. Our magnonic crystals offer large versatility in the design of magnonic band structures and provide a promising avenue towards the realization of energy-efficient magnonic devices.

## Methods

**Sample fabrication**. We grew YIG films with a thickness of 260 nm on GGG(111) substrates using PLD. The GGG substrates were ultrasonically cleaned in acetone and isopropanol before loading into the deposition chamber. We degassed the substrates at 550 °C for 15 min. After this, oxygen was inserted into the chamber and the temperature was raised to 800 °C at a rate of 5 °C per minute. YIG films were deposited from a stoichiometric target in an oxygen partial pressure of 0.13 mbar. We used an excimer laser with a pulse repetition rate of 2 Hz and a laser fluence of 1.8 J cm$^{-2}$. Following film growth, we first annealed the YIG films at 730 °C for 10 min in an oxygen environment of 13 mbar and then cooled them down to room temperature at a rate of −3 °C per minute. The deposition process resulted in single-crystal YIG films, as confirmed by XRD and TEM measurements. The magnonic crystals were fabricated by photolithography. Using a Laserwriter LW405 system, we first defined periodic one-dimensional stripes in a resist layer on top of the 260-nm-thick YIG films. Next, we used argon-ion milling to pattern deep grooves into the YIG film until the GGG substrate was reached. For bicomponent magnonic crystals, this step was followed by deposition of 260 nm $Co_{40}Fe_{40}B_{20}$ using magnetron sputtering at room temperature. Finally, we performed lift-off by placing the samples in a bath of acetone. For spin-wave characterization, parallel microwave antennas with a separation of 200 μm and width of 6 μm were patterned on top of the YIG films. The antennas were fabricated by a similar laser-writing process and consisted of 3 nm Ta and 200 nm Au. Both metals were deposited by magnetron sputtering.

**Broadband spin-wave spectroscopy**. The setup for spin-wave characterization consisted of a two-port VNA (Agilent N5222A) and a home-built electromagnet probing station. We recorded spin-wave transmission spectra by measuring the $S_{12}$ scattering parameter. The excitation power of the microwave signal was set at −10 dBm to avoid nonlinear excitations. We also measured transmission spectra with an excitation power of −30 dBm and, except for stronger background noise, we found identical spin-wave characteristics. We used a frequency sweep method to record spectra for different magnetic bias fields. To improve contrast, we subtracted a reference spectrum taken at 100 mT from imaginary $S_{12}$ data (Figs. 1d, f and 2b, c). The amplitudes of $S_{12}$ shown in Figs. 2b, d–f, 3, 4a–c and 5 are raw data.

**Micromagnetic simulations**. We performed micromagnetic simulations using open-source GPU-accelerated MuMax3 software. The dimensions of the YIG film were set to $x = 655$ μm, $y = 1.5$ μm, and $z = 260$ nm and one-dimensional periodic boundary conditions were applied along the $y$-axis. We discretized the simulations in $40 \times 40 \times 32.5$ nm$^3$ cells. To mimic the experimental magnonic crystals and measurement geometry, we placed $N$ CoFeB stripes or airgaps between 6-μm-wide excitation and detection areas. As input parameters, we used experimentally derived parameters for YIG and CoFeB. An external magnetic bias field was applied along the $y$-axis. For excitation, we used sinc-function-type magnetic field pulses with a cut-off frequency of 3 GHz or sinusoidal ac magnetic fields. In both cases, the field strength was 0.5 mT. During continuous excitation, the time evolution of the $x$-component of magnetization ($m_x$) was recorded for 150 ns. Spatially resolved spin-wave intensity maps were obtained by Fourier-transforming the evolution of $m_x$ on a cell-by-cell basis.

## Data availability

The data that support the findings of this study are available from the corresponding author upon request.

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

## Acknowledgements
This work was supported by the European Research Council (Grant Nos. ERC-2012-StG 307502-E-CONTROL and ERC-PoC-2018 812841-POWERSPIN) and the Academy of Finland (Grant Nos. 317918 and 316857). S.J.H. acknowledges financial support from the Väisälä Foundation. TEM characterization was conducted at the Aalto University Nanomicroscopy Centre (Aalto-NMC) and lithography was performed at the Micronova Nanofabrication Centre, supported by Aalto University. We also acknowledge the computational resources provided by the Aalto Science-IT project.

## Author contributions
H.Q., G.-J.B., and S.v.D. initiated and designed the research. H.Q. fabricated the samples. H.Q. and G.-J.B. conducted the spin-wave measurements. L.Y. carried out TEM characterization. H.Q. and S.J.H. performed the micromagnetic simulations. S.v.D. supervised the project. H.Q. and S.v.D. wrote the manuscript. All authors discussed the results.

## Additional information

**Competing interests:** The authors declare no competing interests.

