## [Peer Review File · Nature Communications]

Reviewers' comments:

Reviewer #1 (Remarks to the Author):

This paper presents microwave measurements and corresponding numerical simulations made on a set of one-dimensional magnonic crystals with a base material of yttrium iron garnet (YIG). While 1D magnonic crystals of YIG have been studied before, this paper represents an advance over the state of the art in several respects. Firstly, the YIG films are significantly thinner than those used in previous work and, as shown in this paper, this allows a signal to propagate even when periodic gaps are inserted that extend all the way through the YIG thickness. Secondly, this paper shows measurements made on bicomponent magnonic crystals where the gaps in the YIG films are filled with CoFeB. The paper shows data for a large number of samples with varied gap widths and gap spacings that provide information on how the magnonic bandwidth and band frequencies depend on these parameters. The microwave data are of high quality and the simulations are an interesting compliment to the data that help to explain some of the advantages that are offered by the thinner YIG and the insertion of the CoFeB. Although the general characteristics of the transmission spectra are similar to what has been shown previously for thicker YIG magnonic crystals, the results are new and the combination of experiment and simulations would be of interest to the community. There are, however, several points that should be addressed that are outlined below.

How significant are the improvements offered by the thinner YIG and the CoFeB? Regarding the statement "these data, we conclude that the separating material (CoFeB versus air) has two main effects. First, bandgaps of the pure YIG magnonic crystal are a bit wider than those of the CoFeB/YIG lattice. Second, transmission of spin waves is more efficient if the airgaps are filled by CoFeB. Both observations agree with the"... These claims would be stronger if quantitative information were added. How big is "a bit"? How big are the improvements in efficiency? Are these changes significant enough to be useful or important? This seems like a central point to the paper and it should be discussed further. According to the data shown in Fig. 3 the effect is minimal when the bars are 2.5 microns wide but larger for bar widths of 5 microns. Were air/CoFeB samples compared for other dimensions and, if so, how do the efficiency improvements and bandgap changes depend on a and w ? Along a similar vein, a good qualitative description of the advantages of thin vs. thick YIG is included but this discussion would benefit from some numbers in order to better frame the advantages of the results from this paper in the context of past work on thicker YIG films.

The authors mention that they needed to add a dead layer in the simulations. "During the course of our simulation study, we noted that good agreements with experimental spin-wave spectra could only be attained with the inclusion of a thin non-magnetic spacer layer. We attribute the formation of this layer to argon-ion milling in our lithography process." How badly did the simulations disagree with the data without the dead layer? Would the performance be enhanced if the dead layer could be reduced or eliminated?

For the simulations, how wide was the magnetic layer? Also, in Fig. 4b, d, e, f was 30 ns chosen (for the green line) because it corresponds to a maximum or is this just an arbitrary time?

The method used for the extraction of the group velocity from the microwave measurements would be improved by some additional explanation. Does this only work where the dispersion curve is linear and is this satisfied for these samples?

In the paper it is mentioned that the effective M_s and the effective anisotropy of the bicomponent materials are extracted from ferromagnetic resonance measurements and then later used for calculations. Are ratios of CoFeB to YIG probed by the FMR measurements the same as in the samples used for the microwave measurements? Are those values really representative of the sample properties?

The plot of Δf vs f in Fig. 1b looks linear but with large intercept. What is the uncertainty in α ? Is this just the Gilbert damping or an effective damping constant that includes other effects, e.g., inhomogeneous broadening? What is the effect of the large intercept on the magnonic crystal performance?

Colorbars should be added to show which colors correspond to positive, negative, or zero.

In figure 3b, there are extra features in the lowest frequency pass band for mid panel?

Reviewer #2 (Remarks to the Author):

The Authors demonstrate experimentally existence of the magnonic band structure with the band gaps in thin one-dimensional magnonic crystals based on stripes of low damping YIG. The transmission of spin waves was enhanced by filling spacers between YIG stripes with CoFeB. Although existing of the band gaps in the spectra of one-dimensional artificial crystals is rather obvious from theoretical point of view, the experimental demonstration with thin low damping YIG has not been demonstrated, so far. Moreover, the Authors pointed at the way for enhancing the coupling strength between magnetization dynamics in YIG stripes. This is very interesting, which can be starting point for further exploitation of spin wave dynamics in magnonic crystals in tunable coupling strength.

The results are interesting and contribute valuable to the magnonics field of research, especially related to exploitation of the periodic structures. In my opinion, the paper can find broad interest in researchers working in the field and make impact on their further development, and can be publishing but the improvement is required.

The experimental demonstration is convincing, however the explanation and theoretical support need improvements. In particular:

- the system measurement system is not clearly described. In Fig. 1c it seems that the SWs are excited by two narrow parts of the CPW, with the microwave field oscillating in opposite directions in both arms. Does it influence the excitation of SWs? Are the SWs excited by the both parts of the CPW with the same phase?
- It is interesting, that the deeper band gaps are obtained already with 4 grooves. Is it revealed in simulations? How does the decrease to only 3 grooves influence the transmission? How will the increase of the number of grooves will modify the spectra.
- The group velocities in periodic structure indicated in the manuscript are not clearly explained. In the one-dimensional model of the periodic structure at the band edges, the group velocity drops to 0, and it is maximal at the miniband middle. How were extracted from the measurements and indicated in the manuscript values and how to read them?
- In the discussion of the influence of the magnetic field orientation on the band structure, there influence of the static demagnetizing field is not mentioned. It seems, that in M perpendicular to the stripes, the strong inhomogeneous demagnetizing field shall exist and influence the spin wave spectra.
- The air gaps between YIG and CoFeB were assumed 150 nm in simulations, while the discretization was 40 nm along the x direction. How does it match?
- If the stripes of YIG and CoFeB are separated, the hysteresis loops for magnetic field along the stripes should indicate the plateau, characteristic for the dipolar coupled systems. Do the Authors measured the hysteresis loop? Did the Authors considered opposite orientation of the magnetization between YIG and CoFeB at some values of the field and its influence on the transmission of SWs?
- The Authors wrote on p. 10 in relation to Fig. 4: "Nodes in spin-wave profiles within the YIG

stripes demonstrate that the spin waves are partially confined." This statement is not clear. If to consider the eigenoscillations in the system, the number of nodes depends on the band number, if the propagating wave at some fixed time is visualized, the position of node depends on the time frame.

- To make the profiles more clear in Fig. 4, I would propose to mark edges of the YIG stripes in Fig. 4, as well as zoom in the profile in CoFeB, show this profile also in the first CoFeB stripe. Perhaps, it could be also informative to plot the x component of the stray magnetic field, through which the coupling between the stripes is assumed to happen.

There are also other minor deficiencies in the manuscript:

- p. 4, in equation defining Δf , the 'c' parameter is not defined or explained its meaning.
- The effect of nonreciprocal excitation was discussed also in other papers: R. Huber, et al., Appl. Phys. Lett. 102, 012403 (2013); V. E. Demidov, et al., Appl. Phys. Lett. 95, 112509 (2009); K. Sekiguchi, et al., Appl. Phys. Lett. 97, 022508 (2010).
- p. 13, there is 'bandstructure' -> 'band structure'.

Reviewer #3 (Remarks to the Author):

The manuscript "Nanometer-thick YIG-based magnonic crystals with large tunable bandgaps" by H. Qin et al. represents interesting experimental studies of spin waves in the YIG-based magnonic crystal. The authors demonstrate that the nanometer-thick YIG-based magnonic crystals allow for large tunable bandgaps of 50 MHz - 200 MHz and the dynamic coupling of separated YIG structures via CoFeB lowers the damping of spin waves. The Authors present interesting results, but however I can not recommend the manuscript for publication in Nature Communications unfortunately.

The magnonic crystals have been widely investigated in the field of magnonics or magnon spintronics. The authors claim that the large bandgaps of 50 MHz - 200 MHz have been observed. However, Z. K. Wang, et al. (ACS Nano 2010 4 (2), 643-648) reported the observed bandgap by BLS measurement over a wide range from 1.4 GHz to 2.6 GHz over eight years ago, which was more than 14 times larger. Moreover, YIG-based magnonic crystals have been substantially investigated by the group of Kaiserslautern (Nat. Commun. 5, 4700 (2014), Appl. Phys. Lett. 106, 102405 (2015), etc.). The authors emphasize that the 270 nm-thick YIG film has the low damping, however, the 20 nm thick YIG films produced by other groups have even lower damping (IEEE Magn. Lett. 5 (2014) 1-4, Appl. Phys. Lett. 110, 092408 (2017) etc.) and are more compatible for the potential magnonic circuits. All in all, the manuscript by H. Qin et al. is not suitable for the publication in Nature Communications, but I would recommend to publish these results in Scientific Reports.

Below are a few technical comments for the consideration of Authors,

- The author compared the spin wave signal strength between the YIG-based magnonic crystals with/without CoFeB, which in my opinion is inappropriate. Considering the infinite damping value of air, the spin wave signal in YIG/CoFeB magnonic crystal is certainly larger.
- The lattice constant of the presented YIG-based magnonic crystal is fairly large (about 30 microns in Fig. 2a) compared to the previous work on magnonic crystals in other materials. Will the bandgap value increase when the lattice constant and the groove width shrink to sub-micrometer size?
- The first word in the title "Nanometer-thick" might be over-emphasized, in my opinion, because there have been plenty of works on 20 nm-thick YIG films since five years ago (APPLIED PHYSICS LETTERS 103, 082408 (2013)) and the film used in this work is 260 nm, i.e. 0.26 micron.

Reviewer #1:

Reviewer comment

This paper presents microwave measurements and corresponding numerical simulations made on a set of one-dimensional magnonic crystals with a base material of yttrium iron garnet (YIG). While 1D magnonic crystals of YIG have been studied before, this paper represents an advance over the state of the art in several respects. Firstly, the YIG films are significantly thinner than those used in previous work and, as shown in this paper, this allows a signal to propagate even when periodic gaps are inserted that extend all the way through the YIG thickness. Secondly, this paper shows measurements made on bicomponent magnonic crystals where the gaps in the YIG films are filled with CoFeB. The paper shows data for a large number of samples with varied gap widths and gap spacings that provide information on how the magnonic bandwidth and band frequencies depend on these parameters. The microwave data are of high quality and the simulations are an interesting compliment to the data that help to explain some of the advantages that are offered by the thinner YIG and the insertion of the CoFeB. Although the general characteristics of the transmission spectra are similar to what has been shown previously for thicker YIG magnonic crystals, the results are new and the combination of experiment and simulations would be of interest to the community. There are, however, several points that should be addressed that are outlined below.

Our response

We would like to thank the reviewer for the assessment of our work. We appreciate his/her observations that our results are new and represent an advance over the state of the art in several respects. Below we will address the reviewer's remarks and questions on a point-by-point basis.

Reviewer comment

How significant are the improvements offered by the thinner YIG and the CoFeB? Regarding the statement "these data, we conclude that the separating material (CoFeB versus air) has two main effects. First, bandgaps of the pure YIG magnonic crystal are a bit wider than those of the CoFeB/YIG lattice. Second, transmission of spin waves is more efficient if the airgaps are filled by CoFeB. Both observations agree with the"... These claims would be stronger if quantitative information were added. How big is "a bit"? How big are the improvements in efficiency? Are these changes significant enough to be useful or important? This seems like a central point to the paper and it should be discussed further. According to the data shown in Fig. 3 the effect is minimal when the bars are 2.5 microns wide but larger for bar widths of 5 microns. Were air/CoFeB samples compared for other dimensions and, if so, how do the efficiency improvements and bandgap changes depend on a and w ? Along a similar vein, a good qualitative description of the advantages of thin vs. thick YIG is included but this discussion would benefit from some numbers in order to better frame the advantages of the results from this paper in the context of past work on thicker YIG films.

Our response

Stimulated by the reviewer's comment, we have added additional experimental data showing the dependence of bandgap size and spin-wave transmission efficiency on bar width w . Results for both CoFeB and air bars are summarized in a new figure (Fig. 4 in the revised manuscript). In this figure, we show transmission spectra for magnonic crystals with lattice parameter $a = 30 \mu\text{m}$ and bar width $w = 2, 4, \text{ and } 6 \mu\text{m}$. The transmission efficiency and bandgap size as a function of w are depicted in Fig. 4d and 4e (these graphs also contain data for $w = 3 \text{ and } 5 \mu\text{m}$). In line with the reviewer's assertion, the difference in transmission efficiency between magnonic crystals with CoFeB and air bars increases with increasing bar width. On the other hand, the bandgap size increase only slightly with w for both types of magnonic

crystals, but their difference remains small. The new data thus corroborate that spin-wave transmission in the allowed minibands can be enhanced substantially by filling the air grooves with CoFeB, while maintaining large bandgaps. The new results of Fig. 4 are discussed on page 8-9 of the revised manuscript (red text). In addition, we have added information regarding the number of scattering units and bandgap size of previously studied YIG-based magnonic crystals to the introduction of the manuscript.

Reviewer comment

The authors mention that they needed to add a dead layer in the simulations. “During the course of our simulation study, we noted that good agreements with experimental spin-wave spectra could only be attained with the inclusion of a thin non-magnetic spacer layer. We attribute the formation of this layer to argon-ion milling in our lithography process.” How badly did the simulations disagree with the data without the dead layer? Would the performance be enhanced if the dead layer could be reduced or eliminated?

Our response

To answer these questions, we have added simulated transmission spectra for a dead layer of 40 nm, 80 nm, 120 nm, 160 nm, and 200 nm to the Supplementary Information (new Supplementary Fig. 5). Dynamic coupling between YIG stripes is stronger if the dead layer is thin. This leads to more narrow and undep bandgaps in the simulated transmission spectra, in contrast to our experimental data. The best agreement between simulations and experiments is attained with a dead layer of 160 nm. If formation of a dead layer would be avoided during the fabrication process, the bandgap would be smaller and less deep. On the other hand, transmission of spin waves in allowed minibands would be more efficient.

Reviewer comment

For the simulations, how wide was the magnetic layer? Also, in Fig. 4b, d, e, f was 30 ns chosen (for the green line) because it corresponds to a maximum or is this just an arbitrary time?

Our response

In the manuscript, we indicate the lattice constant of the magnonic crystal with a and the width of the separating air or CoFeB bars with w . The width of the YIG stripes is thus given by $a - w$. To specify the width of the magnetic layers and air gaps, we use the format $a30w2.5$ in the figures and figure captions. In this example, it means that the magnonic crystal has a lattice constant of 30 μm and its bars (CoFeB or air) are 2.5 μm wide.

The simulations of Fig. 4b, c, e, f (Fig. 6 in the revised manuscript) are performed with a sinusoidal excitation field. It takes about 20 ns for the system to reach a steady excitation state. We therefore show spin-wave profiles at 30 ns. We now explain the reason for choosing this time in the figure caption.

Reviewer comment

The method used for the extraction of the group velocity from the microwave measurements would be improved by some additional explanation. Does this only work where the dispersion curve is linear and is this satisfied for these samples?

Our response

The group velocity of the spin waves (v_g) is extracted from the oscillation period in the $\text{Im}S_{12}$ signal (δt); $v_g = \delta l / \delta t$, with δl being the distance between the two antennas in the spin-wave spectroscopy measurement.

This estimation is accurate if the dispersion curve is approximately linear, which is the case for our YIG films in the relevant frequency range 1.0 – 1.8 GHz (see Fig. 3). To estimate the error due to a small non-linearity in the dispersion curve, we extracted the group velocity at f and $f + \delta f$ at a frequency $f = 1.2$ GHz from the calculated dispersion relation (see Fig. 3). From this analysis, we estimate an error of only 4%. To clarify this, we now write (page 5 of the revised manuscript): “For frequencies ranging from 1.1 GHz to 1.8 GHz, we find a gradual decrease of the group velocity from 11 km/s to 5 km/s (upper panel in Fig. 1e). This method assumes the spin-wave dispersion relation to be linear. The error caused by a small non-linearity in the 1.1 – 1.8 GHz range is estimated as 4%.”

Reviewer comment

In the paper it is mentioned that the effective M_s and the effective anisotropy of the bicomponent materials are extracted from ferromagnetic resonance measurements and then later used for calculations. Are ratios of CoFeB to YIG probed by the FMR measurements the same as in the samples used for the microwave measurements? Are those values really representative of the sample properties?

Our response

The effective M_s and effective anisotropy of the bicomponent materials were extracted from broadband spin-wave spectroscopy measurements, i.e., by recording the S_{12} scattering parameter. These measurements were performed in the same configuration and on the same magnonic crystals with identical CoFeB/YIG ratios as other measurements in the paper. From the spin-wave transmission spectra, we extracted the FMR frequency as a function of external magnetic bias field. The results are summarized in Supplementary Fig. 3. Fits to the data using the Kittel formula give the effective M_s and effective anisotropy of the bicomponent materials. To emphasize that these measurements are performed in transmission and on the same bicomponent magnonic crystals, we now write (page 5 of the revised manuscript): “The values of $\mu_0 H_{\text{ani}}$ and M_s were extracted from the dependence of FMR frequency on external magnetic field (Supplementary Fig. 3). These measurement data were obtained by recording the S_{12} scattering parameter in transmission on the same magnonic crystals.” We also changed the caption of Supplementary Fig. 3 to: “Figure 2: a-c, Kittel-formula fits to experimental FMR data for (a) a continuous YIG film, (b) a a30w2.5 CoFeB/YIG crystal, and (c) a a30w5 CoFeB/YIG crystal. The measurements are performed in transmission. Fitting parameters are given in the graphs.”

Reviewer comment

The plot of Δf vs f in Fig. 1b looks linear but with large intercept. What is the uncertainty in α ? Is this just the Gilbert damping or an effective damping constant that includes other effects, e.g., inhomogeneous broadening? What is the effect of the large intercept on the magnonic crystal performance?

Our response

The linewidth of VNA-FMR spectra as a function of frequency is given by $\Delta f = 2\alpha f + v_g \Delta k$, where α is the Gilbert damping constant, v_g is the spin-wave group velocity, and Δk is the excitation spectrum width of the antenna. Thus, the Gilbert damping constant is extracted from the slope of the linear fit in Fig. 1b (it does not depend on the large intercept, which is caused by the use of a 6 μm -wide antenna). The fit gives $\alpha = (4.4 \pm 1.5) \times 10^{-4}$. We added the uncertainty in the extracted Gilbert damping constant to the manuscript and replaced “ $\Delta f = 2\alpha f + c$ ” with “ $\Delta f = 2\alpha f + v_g \Delta k$ ” to clarify the origin of the intercept.

Reviewer comment

Color bars should be added to show which colors correspond to positive, negative, or zero.

Our response

We added color bars to Fig. 2 and Fig. 4 (Fig. 6 in the revised manuscript).

Reviewer comment

In figure 3b, there are extra features in the lowest frequency pass band for mid panel?

Our response

We investigated the extra feature at about 1.2 GHz in Fig. 3b by performing additional experiments and simulations. The experiments clearly indicate that a suppression of spin-wave transmission in the lowest frequency pass band occurs if the grooves that separate the YIG stripes are wide. Moreover, the suppression is stronger for airgaps than for CoFeB stripes. These observations indicate that the feature is real and that it is more pronounced when dynamic dipolar coupling between the YIG stripes is weak. At the resonance frequency, the wavelength of excited spin waves matches the size of the magnonic crystal (i.e. the distance between the first and last groove). Micromagnetic simulations confirm this effect. The new experimental data and simulations are summarized in a new Supplementary Fig. 4. In the revised manuscript we now write (page 10): “We note the occurrence of an extra feature at ~ 1.2 GHz in the lowest frequency pass band of the a30w4, a30w5, and a30w6 crystals (Figs. 3b and 4b,c). This feature, which signifies a real suppression of the spin-wave transmission signal, appears if the grooves are wide ($w \geq 4$ μm). Moreover, the suppression is stronger if the YIG stripes are separated by air rather than CoFeB. At the resonance frequency, the wavelength of excited spin waves matches the size of the magnonic crystal (i.e. the distance between the first and last groove). For more details see Supplementary Fig. 4.”

Reviewer #2:

Reviewer comment

The Authors demonstrate experimentally existence of the magnonic band structure with the band gaps in thin one-dimensional magnonic crystals based on stripes of low damping YIG. The transmission of spin waves was enhanced by filling spacers between YIG stripes with CoFeB. Although existing of the band gaps in the spectra of one-dimensional artificial crystals is rather obvious from theoretical point of view, the experimental demonstration with thin low damping YIG has not been demonstrated, so far. Moreover, the Authors pointed at the way for enhancing the coupling strength between magnetization dynamics in YIG stipes. This is very interesting, which can be starting point for further exploitation of spin wave dynamics in magnonic crystals in tunable coupling strength.

The results are interesting and contribute valuable to the magnonics field of research, especially related to exploitation of the periodic structures. In my opinion, the paper can find broad interest in researchers working in the field and make impact on their further development, and can be publishing but the improvement is required.

The experimental demonstration is convincing, however the explanation and theoretical support need improvements. In particular:

Our response

We would like to thank the reviewer for the assessment of our work. We appreciate his/her observations that our experimental demonstration is convincing and that our results are interesting and contribute valuable to the magnonics research field. Below we will address the reviewer's questions on a point-by-point basis.

Reviewer comment

- the system measurement system is not clearly described. In Fig. 1c it seems that the SWs are excited by two narrow parts of the CPW, with the microwave field oscillating in opposite directions in both arms. Does it influence the excitation of SWs? Are the SWs excited by the both parts of the CPW with the same phase?

Our response

The reviewer is correct that the microwave field that is generated by the two arms of the antenna are out-of-phase. However, since detection is performed by an identical antenna with two arms, it does not influence the spin-wave transmission measurement. To illustrate this, we compare transmission measurements performed on a continuous YIG film with the same antenna design to measurements where only one of the antenna arms is used (see arXiv:1810.04973). The oscillations in the $\text{Im}S_{12}$ signal and the extracted spin-wave parameters are the same for these two measurements.

Reviewer comment

- It is interesting, that the deeper band gaps are obtained already with 4 groves. Is it revealed in simulations? How does the decrease to only 3 groves influence the transmission? How will the increase of the number of groves will modify the spectra.

Our response

This is an interesting point. It is true that deep and wide bandgaps are readily obtained with only 4 grooves. To assess the influence of groove number on the magnonic band structure, we performed additional experiments. The new data are included as Fig. 5 in the revised manuscript. The results indicate that 2 grooves already produce clear bandgaps. This is totally unprecedented. With increasing groove number, the transmission of spin waves in the allowed minibands increases and the bandgap deepens. The influence of groove number on spin-wave transmission spectra is discussed on page 5 of the revised manuscript. We also performed micromagnetic simulations of magnonic crystals with only two grooves (new Fig. 7). The experiments and simulations agree well. Finally, we added a paragraph to the discussion section (page 16-17) on the very limited number of unit cells in our magnonic crystals.

Reviewer comment

- The group velocities in periodic structure indicated in the manuscript are not clearly explained. In the one-dimensional model of the periodic structure at the band edges, the group velocity drops to 0, and it is maximal at the miniband middle. How were extracted from the measurements and indicated in the manuscript values and how to read them?

Our response

We estimated the group velocity in periodic structures from half oscillation periods in the $\text{Im}S_{12}$ signal (Δf between a maximum and minimum) near the center of the minibands. We clarify this extraction method in the revised version of the manuscript. On page 6, we now write: "From half oscillation periods in the $\text{Im}S_{12}$ signal near the center of the minibands, we derive a spin-wave group velocity ranging from 11 km/s in the first band (~1.2 GHz) to 4.4 km/s in the fourth band (~1.73 GHz). These group velocities are similar to those measured on a continuous YIG film (Fig. 1e)."

Reviewer comment

- The air gaps between YIG and CoFeB were assumed 150 nm in simulations, while the discretization was 40 nm along the x direction. How does it match?

Our response

This is a mistake. The discretization along the x direction was indeed 40 nm, but the airgaps between YIG and CoFeB were 160 nm. We corrected this in the revised manuscript.

Reviewer comment

- If the stripes of YIG and CoFeB are separated, the hysteresis loops for magnetic field along the stripes should indicate the plateau, characteristic for the dipolar coupled systems. Do the Authors measured the hysteresis loop? Did the Authors considered opposite orientation of the magnetization between YIG and CoFeB at some values of the field and its influence on the transmission of SWs?

Our response

We measured the hysteresis loop using magneto-optical Kerr microscopy and indeed observed independent switching in the YIG and CoFeB stripes (see new Supplementary Fig. 1). We also measured spin-wave transmission spectra for parallel and antiparallel magnetization alignments, but did not observe a discernible effect. We attribute this invariance to the large difference in stripe width and the relatively small precession of magnetization in CoFeB at 1.0 – 1.8 GHz. We included this information in the revised

manuscript. On page 7 we now write: “We note that magnetic switching in the YIG and CoFeB stripes occurs independently at a field of about 0.1 mT and 2.0 mT, respectively (Supplementary Fig. 1). An antiparallel magnetization configuration is therefore attained in small bias fields. Judging by the continuous curves of Fig. 2c, switching to a parallel magnetization state at ± 2.0 mT does not affect the spin-wave transmission signal. Reprogramming of the magnonic band structure by toggling between differently aligned magnetization states, as demonstrated for discrete Py wires [11], does therefore not occur in our bicomponent magnonic crystals. We attribute this invariance to the large difference in stripe width and the relatively small precession of magnetization in CoFeB at 1.0 – 1.8 GHz.”

Reviewer comment

- The Authors wrote on p. 10 in relation to Fig. 4: “Nodes in spin-wave profiles within the YIG stripes demonstrate that the spin waves are partially confined.” This statement is not clear. If to consider the eigenoscillations in the system, the number of nodes depends on the band number, if the propagating wave at some fixed time is visualized, the position of node depends on the time frame.

Our response

The simulations in Fig. 4a,d (Fig. 6a,d in the revised manuscript) are performed by pulse excitation (we use a sinc-function-type magnetic field pulse with a cut-off frequency of 3 GHz) and Fourier transforming of the evolution of m_x on a cell-by-cell basis. The result is a measure of the spin-wave intensity as a function of frequency and position. The reviewer correctly points out that this is not the same as the spin-wave profile. We have corrected this. The simulations in Fig. 4b,c,e,f (Fig. 6b,c,e,f in the revised manuscript) are performed by continuous excitation with a sinusoidal ac magnetic field at a fixed frequency. After 20 ns, a steady-state is reached. After this, the spin-wave profile depends on the time-frame, but the positions of the nodes in the YIG stripes do not change (partially confined modes). In Fig. 6b,e, we plot spin-wave profiles for a fixed excitation frequency of 1.44 GHz. At this frequency, one node in the middle of the first YIG stripes can be seen. In the caption of Fig. 6, we now distinguish between the types of data shown in a,d and b,c,e,f. We also clarify the different excitation mechanisms in these simulations.

Reviewer comment

- To make the profiles more clear in Fig. 4, I would propose to mark edges of the YIG stripes in Fig. 4, as well as zoom in the profile in CoFeB, show this profile also in the first CoFeB stripe. Perhaps, it could be also informative to plot the x component of the stray magnetic field, through which the coupling between the stripes is assumed to happen.

Our response

We appreciate the reviewer’s suggestion to plot the x component of the dynamic dipolar field between the YIG stripes. To do this, we performed additional simulations on magnonic crystals with a lattice period of 30 μm and groove widths of 2.5 μm and 5 μm . In these simulations, the groove width is fixed, but the amount of CoFeB inside the grooves is systematically varied from 0 (pure airgap) to the width of the groove. Results for the a30w2.5 magnonic crystal are shown in Fig. 6g (bottom panel). The data illustrate that the amplitude of the dynamic dipolar field increases with the width of the CoFeB stripe inside the groove. Figure 6h summarizes the results for both magnonic crystals. The new simulations confirm that filling of the grooves with CoFeB enhances the dynamic coupling between discrete YIG stripes.

In the new version of Fig. 4 (Fig. 6 in the revised manuscript), we now mark the edges of the YIG stripes in b,c,e,f with dashed red lines. We also present zoom-ins of the spin-wave amplitude near the first CoFeB stripe (top panel of Fig. 6g).

Reviewer comment

There are also other minor deficiencies in the manuscript:

- p. 4, in equation defining Δf , the 'c' parameter is not defined or explained its meaning.

Our response

We replaced " $\Delta f = 2\alpha f + c$ " with " $\Delta f = 2\alpha f + v_g \Delta k$ " to clarify the origin of the intercept.

Reviewer comment

- The effect of nonreciprocal excitation was discussed also in other papers: R. Huber, et al., Appl. Phys. Lett. 102, 012403 (2013); V. E. Demidov, et al., Appl. Phys. Lett. 95, 112509 (2009); K. Sekiguchi, et al., Appl. Phys. Lett. 97, 022508 (2010).

Our response

We added the suggested references to the manuscript.

Reviewer comment

- p. 13, there is 'bandstructure' -> 'band structure'.

Our response

We corrected the spelling of "band structure".

Reviewer #3:

Reviewer comment

The manuscript "Nanometer-thick YIG-based magnonic crystals with large tunable bandgaps" by H. Qin et al. represents interesting experimental studies of spin waves in the YIG-based magnonic crystal. The authors demonstrate that the nanometer-thick YIG-based magnonic crystals allow for large tunable bandgaps of 50 MHz - 200 MHz and the dynamic coupling of separated YIG structures via CoFeB lowers the damping of spin waves. The Authors present interesting results, but however I can not recommend the manuscript for publication in Nature Communications unfortunately. The magnonic crystals have been widely investigated in the field of magnonics or magnon spintronics. The authors claim that the large bandgaps of 50 MHz - 200 MHz have been observed. However, Z. K. Wang, et al. (ACS Nano 2010 4 (2), 643-648) reported the observed bandgap by BLS measurement over a wide range from 1.4 GHz to 2.6 GHz over eight years ago, which was more than 14 times larger. Moreover, YIG-based magnonic crystals have been substantially investigated by the group of Kaiserslautern (Nat. Commun. 5, 4700 (2014), Appl. Phys. Lett. 106, 102405 (2015), etc.). The authors emphasize that the 270 nm-thick YIG film has the low damping, however, the 20 nm thick YIG films produced by other groups have even lower damping (IEEE Magn. Lett. 5 (2014) 1-4, Appl. Phys. Lett. 110, 092408 (2017) etc.) and are more compatible for the potential magnonic circuits. All in all, the manuscript by H. Qin et al. is not suitable for the publication in Nature Communications, but I would recommend to publish these results in Scientific Reports.

Our response

We would like to thank the reviewer for the assessment of our work. He/she finds our results interesting and compares our work to previous studies on magnonic crystals. First the reviewer states that Z. K. Wang et al. (ACS Nano 4, 643-648 (2010)) reported on magnonic crystals having a 14 times larger band gap (we reference to a similar publication by Wang et al. in our manuscript (Ref. 9 in the original manuscript)). We would like to point out that the results by Z. K. Wang et al. were attained in magnonic crystals made of Co or Py. Other groups have also demonstrated large bandgaps using ferromagnetic metals. The reason for large bandgaps in these magnonic crystals is the large saturation magnetization of the constituting materials. As described in the introduction of our manuscript, the big drawback of ferromagnetic metals is strong magnetic damping, which limits the transmission of spin waves. YIG offers about two orders of lower magnetic damping and, thus, is a pivotal material for magnonics. However, the saturation magnetization of YIG is much smaller than that of ferromagnetic metals and, consequently, spin waves in YIG are less dispersive. As a result, it is much more difficult to open up large bandgaps in YIG-based magnonic crystals. Up to now, the magnonic bandgap of such crystals ranges from only a few to tens of MHz. In our work, we show that the bandgap of YIG-based magnonic crystals can be substantially enhanced by a reduction of the film thickness (from several micrometers in previous studies to 260 nm) and patterning of physically separated YIG stripes. The main effect of thickness reduction is an increase of the depolarization field. As a consequence, magnetization precessions in the out-of-plane directions are effectively suppressed. Because in-plane and out-of-plane magnetization precessions favor in-phase and out-of-phase oscillations in neighboring stripes, respectively, spin-wave transport is substantially suppressed if their dipolar coupling fields are comparable (this is the case in thick magnonic crystals). In our nanometer-thick YIG-based magnonic crystals, the in-plane dynamic dipolar field dominates, allowing for efficient spin-wave transmission. Because of this, we still attain good transmission when we fully separate the YIG stripes by airgaps or CoFeB. In previous studies on thicker YIG-based magnonic crystal, shallow grooves were used to avoid complete suppression of the spin-wave signal. Bragg reflection by shallow grooves is not efficient. Stronger Bragg reflection by the airgaps or narrow CoFeB stripes that fully extend throughout the YIG film in our study has several striking consequences: (1) It leads to the formation of larger bandgaps (the size of the bandgap is defined by the reflection efficiency of an individual reflector) compared to other studies on YIG-based magnonic crystals. (2) Only a very limited number of units is

required to open up deep bandgaps. In the revised manuscript, we present new experimental (Fig. 5) and simulation (Fig. 7) data demonstrating that only two airgaps or CoFeB stripes in nanometer-thick YIG films already produce clear bandgaps. This is fully unprecedented. In previous studies, typically 10 – 20 unit cells are needed to sufficiently suppress spin-wave transmission at the bandgap frequency. (3) Because of efficient Bragg scattering and low damping in YIG, up to 7 bandgaps are resolved in crystals with $a \geq 50 \mu\text{m}$.

The reviewer cites work on YIG magnonic crystals by the group of Kaiserslautern. In our paper, we extensively reference to papers by this group (Refs. 1,7,14,15,16,21 in the original manuscript). In Refs. 15 and 16, which specifically report on YIG-based magnonic crystals with grooves, the thickness of the crystals is 5.5 micrometers, the grooves are shallow, and the magnonic bandgaps are small, ranging from 10 to 30 MHz. In these works, 10 grooves are used to open up bandgaps because of relatively inefficient Bragg scattering by the shallow grooves (see discussion above). The first paper cited by the reviewer (Nat. Commun. 5, 4700 (2014)) reports on a magnon transistor that utilizes a YIG-based magnonic crystal. Again, the crystal is 5.5 micrometer thick and it consists of 10 shallow grooves, producing a similar magnonic band structure as in Refs. 15 and 16. The second paper cited by the reviewer (Appl. Phys. Lett. 106, 102405 (2015)) reports on a spin-wave logic gate that is based on a width-modulated dynamic magnonic crystal. The design of this magnonic crystal is different, it consists of 10 units, and its bandgap is about 10 MHz. In all cases, the magnonic bandgaps are substantially smaller and the number of units is much larger than in our work on nanometer-thick and physically separated YIG stripes. Again, the main reason for the good performance of our thin magnonic crystal is a reduction of out-of-plane magnetization precessions in the YIG stripes. Because of this, the YIG stripes predominantly interact via in-plane dynamic dipolar fields that favor in-phase magnetization oscillations. We are therefore able to physically separate the YIG stripes while still retaining good spin-wave transmission, something that is impossible in previously studied micrometer-thick YIG-based magnonic crystals.

In response to comments by the reviewer, we added the following references to our manuscript: ACS Nano 4, 643-648 (2010), Nat. Commun. 5, 4700 (2014), and Appl. Phys. Lett. 106, 102405 (2015). We also added data demonstrating bandgap formation by only two airgaps or narrow CoFeB stripes in nanometer-thick YIG films (new Figs. 5 and 7).

Finally, the reviewer comments on the damping constant of our YIG films. Continuous YIG films with low damping constants have become available recently and we agree that this is a promising development for the magnonics research field. However, the novelty of our paper is not related to the damping parameter of our YIG film. Instead, we for the first time demonstrate versatile bandgap engineering in nanometer-thick YIG-based magnonic crystals.

Reviewer comment

Below are a few technical comments for the consideration of Authors.

- The author compared the spin wave signal strength between the YIG-based magnonic crystals with/without CoFeB, which in my opinion is inappropriate. Considering the infinite damping value of air, the spin wave signal in YIG/CoFeB magnonic crystal is certainly larger.

Our response

In the revised manuscript, we systematically quantify the effect of CoFeB on the efficiency of spin-wave transmission. Using micromagnetic simulations, we show that the dynamic dipolar field increases with the amount of CoFeB in the air grooves (new Figs. 6g,h). The enhancement of the coupling field is caused by forced precessions of magnetization in CoFeB. While the precessions are small at 1.0 – 1.8 GHz, they produce a significant dipolar field because the magnetization of CoFeB is large. The dynamic field that the CoFeB stripes produce adds to the dipolar field from quasi-standing modes in the YIG stripes and,

consequently, it promotes spin-wave propagation across the magnonic crystal. Enhanced stripe-to-stripe coupling also affects the band structure. Near-perfect confinement of spin-wave modes in the stripes of a one-dimensional crystal with parameters w (gap width) and a (lattice constant) would in theory produce narrow minibands for spin waves with $k=n\pi/(a-w)$ and these bands would be separated by large frequency gaps. Dynamic dipolar coupling between stripes reduces the confinement of spin-wave modes. For constant w and a , an increase of the coupling strength would thus enhance the miniband width and reduce the bandgap size. This is exactly what we observe in our experiments (Fig. 3 and new Fig. 4) and simulations (Fig. 6 and new Fig. 7) when the air grooves of our YIG-based magnonic crystals are filled by CoFeB.

Reviewer comment

- The lattice constant of the presented YIG-based magnonic crystal is fairly large (about 30 microns in Fig. 2a) compared to the previous work on magnonic crystals in other materials. Will the bandgap value increase when the lattice constant and the groove width shrink to sub-micrometer size?

Our response

Data in Fig. 8 of the revised manuscript show that the bandgap increases with decreasing lattice constant and increasing groove width. We simulated the effect of a further reduction of the lattice constant towards sub-micromagnetic sizes (see figure below). As one can see, the bandgap increase further with decreasing lattice constant, but at some point it starts to decrease because of a flattening of the spin-wave dispersion curve (c).

Micromagnetic simulations of the bandgap size and frequency of magnonic crystals with reducing lattice constants and groove widths. The spin-wave dispersion relation and wave vector corresponding to π/a are shown in (c).

Reviewer comment

- The first word in the title "Nanometer-thick" might be over-emphasized, in my opinion, because there have been plenty of works on 20 nm-thick YIG films since five years ago (APPLIED PHYSICS LETTERS 103, 082408 (2013)) and the film used in this work is 260 nm, i.e. 0.26 micron.

Our response

We emphasize that we do not report on continuous YIG films such as those reported in Applied Physics Letters 103, 082408 (2013) and other papers, but on the first nanometer-thick YIG based magnonic crystal (as clearly stated in the title). Thickness reduction plays a critical role in the opening of large bandgaps in our YIG-based magnonic crystals with only $N = 2 - 4$ units (see response to the first comment). We therefore find it appropriate to us “nanometer-thick YIG-based magnonic crystals” in the title.

Reviewers' comments:

Reviewer #1 (Remarks to the Author):

The revised version is improved over the original and most questions are addressed adequately. There are, however, a few points that would still need to be addressed prior to publication.

Regarding the description of the extraction of the group velocity. This part is improved but is still confusing, particularly in the sentence on p. 5 that starts "The group velocity of the spin waves (v_g) is extracted from the oscillation period (Δf) ...". A period is usually taken to be the time it takes for one oscillation (i.e., the inverse of the frequency). That is not what is meant here and the mismatch of the units (time is expected but Δf has units of frequency) is likely to be confusing to the reader. Add a reference to Fig. 1d to explain what is meant by Δf and revise the description to make this more understandable.

The discussion on the efficiency still needs to be made more quantitative. For example, on p. 9 "The results clearly demonstrate that the efficiency of spinwave transmission drops substantially with air-groove width. In contrast, the amplitude of the S12 signal depends much less on w if the grooves are filled with CoFeB." The phrases "drops substantially", "depends much less on w ..." should be made more quantitative.

Now that there are more figures in the supplemental materials, some (brief) text should be added to the supplemental materials to better explain the importance of the figures.

Reviewer #2 (Remarks to the Author):

I have found satisfactory answers for the Reviewers comments and the valuable improvement of the manuscript, with detailed explanation of new effects which appear in magnonic crystals under investigation. Thus, the manuscript is ready for publication in my opinion.

Reviewer #3 (Remarks to the Author):

In the revised version of manuscript and Supplementary material, the authors did reply to all the comments made by reviewers. The authors put additional experimental data showing the dependence of band gap size and spin-wave transmission efficiency on air and CoFeB bar width and numbers. They also added simulation spectra for a dead layer of different thickness in Supplementary Information. The explanation and theoretical part are improved as well.

Although all efforts mentioned above are well appreciated, I still think this work is not yet suitable to be published in Nature Communications. The experimental work investigated in this manuscript is far from cutting-edge nanotechnology. The authors state in the response that they used fully extended grooves throughout the YIG film that has striking consequences than previous studies. The authors choose to compare with the results to the YIG film with shallow grooves and make the expression of large band gap. In fact, the isolated YIG stripes of fully extended grooves are not too different from antidote (2D) of permalloy (Appl. Phys. Lett. 106, 262406 (2015)). If one compares with Appl. Phys. Lett. 106, 262406 (2015), their period or lattice constant is about 300 nm that is two orders of magnitude smaller than what is investigated in this work 30 microns. Concerning the YIG filled with CoFeB, Grundler's group has successfully fabricated device with 20-nm thick YIG film, etched and filled with CoFeB. (Nat. Commun. 7, 11255 (2016)) with period down to 600 nm, although this is 2D structure. YIG stripes with Cobalt bars are also similar to Py/Co bicomponent magnonic crystals (ACS Nano 4, 643-648 (2010)). For thin film YIG based one dimensional structure, strong coupling between Co bars and YIG film (Phys. Rev. Lett. 120.217202 (2018)) are

observed lately, in which the large anticrossing gap can also be interpreted as large band gap in the dispersion relation. Why are isolated YIG stripes and YIG/Co bicomponent stripe structures more striking than Py stripes (Appl. Phys. Lett. 100, 073114 (2012)) and Py/Co structures? These results are in fact quite expected based on the large number of experimental works in the last decade. (see e.g., review J. Phys.: Condens. Matter 26 (2014) 123202).

I fully agree that it is only recently (circa 2013) that the nanometer-thick high quality YIG film becomes available that powers the research on nm-thick YIG film with nanostructures, such as this work. Group at Colorado state university did some pioneer works on this (IEEE Magn. Lett. 5, 6700104 (2014).) and deserve to be mentioned.

As for the title of the manuscript, I still do not think 260 nm can be legitimately called "nanometer". It is too vague, since people working on 20 nm even 10 nm thick YIG also call it nm-thick. That would be unfair for them. I would put it as "sub-micron".

A remarkable greatness of this experimental work is that a strong transmission signal is clearly observed after propagation of 200 microns. This is indeed spectacular, since although the damping of nm-thick YIG film is estimated to be 10^{-4} , the real decay length is still of controversy. This long-distance propagation (10 microns for Nat. Phys. 11, 1022–1026 (2015)). of spin waves should be appreciated and demonstrate a exceptionally high quality of the YIG film. The authors even should consider put "long-distance" in the title, since till now, most of the magnonic crystal works are studied by spin wave "reflection" method, not long-distance transmission.

In principle, I feel the work investigated here is interesting and done in high quality. However, the nanostructure fabricated here is not yet pushed to the limit of the cutting-edge nanotechnology. If the authors may work on either a thinner film, below 100 nm (in the title one can then put nanometer-thick) OR with period below 1 micron that nanostructures are investigated, this work will deserve to be published in Nature Communications. Last but not the least, previous works and pioneer works are suggested to be mentioned in the revised version.

Reviewer #1:

Reviewer comment

The revised version is improved over the original and most questions are addressed adequately. There are, however, a few points that would still need to be addressed prior to publication.

Regarding the description of the extraction of the group velocity. This part is improved but is still confusing, particularly in the sentence on p. 5 that starts "The group velocity of the spin waves (v_g) is extracted from the oscillation period (δf) ...". A period is usually taken to be the time it takes for one oscillation (i.e., the inverse of the frequency). That is not what is meant here and the mismatch of the units (time is expected but δf has units of frequency) is likely to be confusing to the reader. Add a reference to Fig. 1d to explain what is meant by δf and revise the description to make this more understandable.

Our response

We would like to thank the reviewer for the assessment of our work. We appreciate that he/she finds the revised version improved and recommends publication after we address a few more points.

Regarding the description of the extraction of the group velocity, we agree with the reviewer that it may be confusing to talk about oscillation period in the frequency domain. We have modified the description and added a reference to Fig. 1d. The text now reads: "The group velocity of the spin waves (v_g) is extracted from the frequency separation (δf , indicated in Fig. 1d) using $v_g = \delta f \times x$ [29,30]."

Reviewer comment

The discussion on the efficiency still needs to be made more quantitative. For example, on p. 9 "The results clearly demonstrate that the efficiency of spinwave transmission drops substantially with air-groove width. In contrast, the amplitude of the S_{12} signal depends much less on w if the grooves are filled with CoFeB." The phrases "drops substantially", "depends much less on w ..." should be made more quantitative.

Our response

We made the sentences on p. 9 more quantitative. It now reads: "The results demonstrate that the efficiency of spin-wave transmission drops 16 and 22 times in the 2nd and 3rd miniband if the air-groove width is enlarged from 2 μm to 6 μm . In contrast, the S_{12} signal only reduces by a factor <3 upon an increase of w if the grooves are filled with CoFeB. Because of these dissimilar dependencies, the transmission of spin waves in higher order minibands of CoFeB/YIG lattices is at least one order of magnitude larger than in crystals made of pure YIG if $w > 5 \mu\text{m}$."

Reviewer comment

Now that there are more figures in the supplemental materials, some (brief) text should be added to the supplemental materials to better explain the importance of the figures.

Our response

We added some brief text to the supplementary information and the supplementary figures to better explain their importance.

Reviewer #2:

Reviewer comment

I have found satisfactory answers for the Reviewers comments and the valuable improvement of the manuscript, with detailed explanation of new effects which appear in magnonic crystals under investigation. Thus, the manuscript is ready for publication I my opinion.

Our response

We would like to thank the reviewer for his/her positive comments on the new effects that appear in our magnonic crystals and the recommendation to publish.

Reviewer #3:

Reviewer comment

In the revised version of manuscript and Supplementary material, the authors did reply to all the comments made by reviewers. The authors put additional experimental data showing the dependence of band gap size and spin-wave transmission efficiency on air and CoFeB bar width and numbers. They also added simulation spectra for a dead layer of different thickness in Supplementary Information. The explanation and theoretical part are improved as well.

Our response

We note that we also included experimental and simulation data on the dependence of bandgap formation and spin-wave transmission on the number of grooves. According to these new results, robust bandgaps are already produced by two grooves, a feature that has never been demonstrated before.

Reviewer comment

Although all efforts mentioned above are well appreciated, I still think this work is not yet suitable to be published in Nature Communications. The experimental work investigated in this manuscript is far from cutting-edge nanotechnology. The authors state in the response that they used fully extended grooves throughout the YIG film that has striking consequences than previous studies. The authors choose to compare with the results to the YIG film with shallow grooves and make the expression of large band gap. In fact, the isolated YIG stripes of fully extended grooves are not too different from antidote (2D) of permalloy (Appl. Phys. Lett. 106, 262406 (2015)). If one compares with Appl. Phys. Lett. 106, 262406 (2015), their period or lattice constant is about 300 nm that is two orders of magnitude smaller than what is investigated in this work 30 microns. Concerning the YIG filled with CoFeB, Grundler's group has successfully fabricated device with 20-nm thick YIG film, etched and filled with CoFeB. (Nat. Commun. 7, 11255 (2016)) with period down to 600 nm, although this is 2D structure. YIG stripes with Cobalt bars are also similar to Py/Co bicomponent magnonic crystals (ACS Nano 4, 643-648 (2010)). For thin film YIG based one dimensional structure, strong coupling between Co bars and YIG film (Phys. Rev. Lett. 120.217202 (2018)) are observed lately, in which the large anticrossing gap can also be interpreted as large band gap in the dispersion relation. Why are isolated YIG stripes and YIG/Co bicomponent stripe structures more striking than Py stripes (Appl. Phys. Lett. 100, 073114 (2012)) and Py/Co structures? These results are in fact quite expected based on the large number of experimental works in the last decade. (see e.g., review J. Phys.: Condens. Matter 26 (2014) 123202).

Our response

The reviewer focuses on the thickness and lattice constant of magnonic crystals rather than their physical properties. In our manuscript we report on discrete YIG-based magnonic crystals with unique features, including low-loss and long-distance spin wave propagation, a tenfold increase of the bandgap size compared to previous studies on low-loss YIG, measurements of up to 7 bandgaps, and the first demonstration of robust bandgap formation with only two scattering units.

As outlined in the introduction, YIG offers much lower magnetic damping than ferromagnetic metals, but its small saturation magnetization complicates the opening of magnonic bandgaps because of weak spin-wave dispersion. Our work addresses this challenge. We show that dynamic dipolar coupling between discrete stripes becomes large enough for efficient spin-wave transmission when the thickness of YIG is reduced to sub-micrometer levels. Because we pattern the grooves all the way to the substrate, Bragg scattering on individual units is highly effective. As a result, an unprecedented combination of efficient and long-distance spin-wave propagation in allowed minibands and strong suppression of spin-wave

transmission in robust bandgaps is obtained with only two scattering units. We stress that similar results have not been reported in other YIG-based magnonic crystals or crystals made from ferromagnetic metals.

In his/her previous report, the reviewer requested the inclusion of additional references on magnonic crystals made of YIG (thick films with shallow grooves) and ferromagnetic metals. In response, we added three references. Now the reviewer is requesting other references to be added. First the reviewer discusses work on antidot lattices made of permalloy. We are well aware of these works but like to point out that the challenges of opening up bandgaps in YIG-based magnonic crystals are very different from that in crystals made of ferromagnetic metals. The much larger saturation magnetization of ferromagnetic metals produces a strong spin-wave dispersion. Consequently, flattening of the dispersion relation in nanometer thick films is not problematic for these materials as the dispersion remains large enough for versatile band structure engineering. Many works on magnonic crystals made of ferromagnetic metals have been published where the film and lattice period are on the nanometer scale. References 8 – 14 in our manuscript highlight some of these works. Review articles (Refs. 2-7) provide further details on achievements in the field. As stated before, the challenges in YIG-based magnonic crystals are different. Because of a much smaller saturation magnetization, the spin-wave dispersion relation is relatively flat and dynamic coupling between discrete elements is weak. To overcome these drawbacks, all published works on YIG-based magnonic crystals thus far focused on thick films with relatively weak scattering units (e.g. shallow grooves). Our results significantly advance the properties of YIG-based magnonic crystals, which is an important step towards the utilization of low-loss YIG-based metamaterials.

The reviewer also cites work by the Grundler group on a device with 20-nm thick YIG film, etched and filled with CoFeB. (Nat. Commun. 7, 11255 (2016)). The reviewer only mentions the thickness parameter but does not elaborate on the physics of the device. In fact, the interesting paper by the Grundler group reports on short-wavelength spin-wave emission via a grating coupler effect. It does not report on a YIG-based magnonic crystal. As a side note, the YIG film in this work is not patterned on the nanoscale. The CoFeB does therefore not fill holes in the YIG films, but is patterned on top. Contrary to ferromagnetic metals, patterning of insulating YIG films on the nanoscale is very challenging.

Next, the reviewer writes “YIG stripes with Cobalt bars are also similar to Py/Co bicomponent magnonic crystals (ACS Nano 4, 643-648 (2010)). Again a comparison with ferromagnetic metals is made. References 8-14 in our manuscript include discussions on bicomponent metal crystals. As explained above, the challenges for magnonic crystal made of ferromagnetic metals and ultralow damping YIG are different. We already added the article ACS Nano 4, 643-648 (2010) in response to the reviewer’s first report (Ref. 14).

The reviewer mentions coupling between Co bars and a YIG film (Phys. Rev. Lett.120.217202 (2018)). This work reports on hybridization between FMR modes in YIG and Co causing anti-crossing behavior in FMR spectra. This anti-crossing behavior should not be confused with the opening of a bandgap in spin-wave transmission spectra. In fact, the cited paper does not contain any transmission data. We recently published a similar paper on coupling between FMR modes in CoFeB and YIG, leading to the excitation of perpendicular standing spin waves (Sci. Rep. 8, 5755 (2018)). Because the physics of mode coupling in such bilayers is very different from band structure engineering in magnonic crystals, we did not cite this work.

To answer the final question about why our results are so striking compared to previous work on Py and Py/Co structures (i.e. ferromagnetic metals), we summarize the main conclusions of our study:

- Our YIG-based magnonic crystals exhibit low-loss and long-distance spin-wave propagation in allowed minibands. This property is superior to ferromagnetic metals. We added some text to emphasize this more (see response to comments below).
- Because of low magnetic damping and efficient scattering on individual lattice units, only 2 grooves are required for the formation of robust bandgaps. This is totally unprecedented as previous YIG-based *and* ferromagnetic metal structures typically require >5 units.

- The bandgap size is very large compared to previous experiments on YIG-based magnonic crystals (our bandgap is one order of magnitude wider than the largest values previously reported). The reason for the large bandgap is a reduction of film thickness, which enhances dynamic dipolar coupling and allows the grooves to be extended to the substrate. Because of the fully discrete nature of our YIG-based magnonic crystal (the first ever reported), Bragg scattering on individual units is drastically enhanced and the bandgap is larger and deeper.
- Because of low damping and efficient scattering, we measure up to 7 bandgaps.
- We show that filling of the grooves between YIG stripes with CoFeB significantly enhances the transmission of spin waves even further with only a minor reduction of the bandgap width.

All the above conclusions are corroborated by detailed measurements on magnonic crystals with different groove width, lattice parameter, and groove number. Micromagnetic simulations confirm the results and provide insights on the underlying physics. When considering the physics of our magnonic crystals, we hope that the reviewer agrees with reviewer 1 and 2 that “our results are new, represent an advance over the state of the art in several respects, and contribute valuable to the magnonics research field.”

Finally, we note that article Appl. Phys. Lett. 100, 073114 (2012) reports on magnetic logic based on a meander-type Py nanowire. The implementation of this structure is different and, more importantly, it does not exhibit any of the features listed above. We already cite 4 papers of this group (Refs. 9, 10, 11, and 14, Ref. 14 was added after the reviewer suggested it in his/her previous report). While we do not think that another citation to similar work is needed, we added a reference to Appl. Phys. Lett. 100, 073114 (2012) in the revised version of manuscript (Ref. 15).

Reviewer comment

I fully agree that it is only recently (circa 2013) that the nanometer-thick high quality YIG film becomes available that powers the research on nm-thick YIG film with nanostructures, such as this work. Group at Colorado state university did some pioneer works on this (IEEE Magn. Lett. 5, 6700104 (2014).) and deserve to be mentioned.

Our response

In our manuscript we already reference to work on the growth of high-quality YIG films by the group at Colorado State University (Ref. 39).

Reviewer comment

As for the title of the manuscript, I still do not think 260 nm can be legitimately called “nanometer”. It is too vague, since people working on 20 nm even 10 nm thick YIG also call it nm-thick. That would be unfair for them. I would put it as “sub-micron”.

Our response

We removed “nanometer-thick” from the title of the manuscript.

Reviewer comment

A remarkable greatness of this experimental work is that a strong transmission signal is clearly observed after propagation of 200 microns. This is indeed spectacular, since although the damping of nm-thick YIG film is estimated to be 10^{-4} , the real decay length is still of controversy. This long-distance propagation (10 microns for Nat. Phys. 11, 1022–1026 (2015)). of spin waves should be appreciated and demonstrate an exceptionally high quality of the YIG film. The authors even should consider put “long-distance” in the title, since till now, most of the magnonic crystal works are studied by spin wave “reflection” method, not long-

distance

transmission.

Our response

Low-loss transmission of spin waves in our YIG-based magnonic crystals is indeed one its attractive properties. Compared to ferromagnetic metals where the decay length of spin waves does not exceed a few tens of micrometers, we measure efficient spin-wave transmission between antennas that are separated by 200 micrometers. Extrapolation of the data shows that spin-wave propagation over distances exceeding 1 millimeter are possible. Besides this great feature, ultralow damping in YIG also enables a reduction of the number of scattering units (grooves) to the fundamental limit of 2. The effect of damping on the required number of scattering units is nicely illustrated by the micromagnetic simulations of Supplementary Fig. 6b. To emphasize low-loss transmission in our YIG-based magnonic crystals we changed the title of the manuscript to: "Low-loss YIG-based magnonic crystals with large tunable bandgaps." We also included the following text to the introduction of the manuscript: "However, magnonic crystals made of ferromagnetic metals suffer from inefficient transmission because of strong Gilbert damping. The decay length of spin waves in these materials is limited to a few tens of micrometers [7]" and later: "Here, we experimentally demonstrate discrete YIG-based magnonic crystals combining robust tunable bandgaps of up to 200 MHz and low-loss transmission in allowed minibands over a distance of 200 μm ."

Reviewer comment

In principle, I feel the work investigated here is interesting and done in high quality. However, the nanostructure fabricated here is not yet pushed to the limit of the cutting-edge nanotechnology. If the authors may work on either a thinner film, below 100 nm (in the title one can then put nanometer-thick) OR with period below 1 micron that nanostructures are investigated, this work will deserve to be published in Nature Communications. Last but not the least, previous works and pioneer works are suggested to be mentioned in the revised version.

Our response

With respect to the last comment, we extensively reference to previous works on magnonic crystals made of ferromagnetic metals (Refs. 8-15) and YIG (Refs. 17-26). Moreover we cite 7 review articles on magnonic crystals and YIG magnonics (Refs. 2-7 and 16), providing a detailed overview of the field. In response to the reviewer, we honored all four reference suggestions to articles on magnonic crystals made of YIG (Refs. 20 and 25) and ferromagnetic metal stripes (Refs. 14 and 15).

In response to the first report of the reviewer, we performed additional simulations on the influence of lattice constant on the properties of YIG-based magnonic crystals. These data, which were presented in our response letter, indicate that the enlargement of the bandgap starts to saturate when the lattice constant becomes small. A further reduction of the YIG film thickness is expected to lower the decay length and change the magnonic band structure in a complex way. Importantly, scaling on the nanoscale in YIG-based magnonic crystals cannot be extrapolated from previous works on ferromagnetic metals because of its weaker spin-wave dispersion and saturation magnetization. This is an interesting optimization problem. Our results provide clear handles on how to improve the properties of YIG-based magnonic crystals (thickness reduction and discretization) and will therefore stimulate further research in this direction. One of the challenges towards this goal is the patterning of insulating high-quality YIG films on the nanoscale. Because our results already go well beyond the current state-of-the-art in low-loss YIG-based magnonics (including a tenfold increase of the bandgap, the first demonstration of a robust magnonic crystal with only two scattering units, and efficient long-distance propagation of spin waves in allowed minibands) and provide clear directions for further research, we believe that our work deserves to be published in Nature Communications.